# General Exploratory Bonus for Optimistic Exploration in RLHF

**Wendi Li, Changdae Oh, Sharon Li**
Department of Computer Sciences
University of Wisconsin-Madison
{wli679,changdae,sharonli}@cs.wisc.edu

## Abstract

Optimistic exploration is central to improving sample efficiency in reinforcement learning with human feedback, yet existing exploratory bonus methods to incentivize exploration often fail to realize optimism. We provide a theoretical analysis showing that current formulations, under KL or $\alpha$-divergence regularization, unintentionally bias exploration toward high-probability regions of the reference model, thereby reinforcing conservative behavior instead of promoting discovery of uncertain regions. To address this pitfall, we introduce the **General Exploratory Bonus** (**GEB**), a novel theoretical framework that provably satisfies the optimism principle. GEB counteracts divergence-induced bias via reference-dependent reward regulation and unifies prior heuristic bonuses as special cases, while extending naturally across the full $\alpha$-divergence family. Empirically, GEB consistently outperforms baselines on alignment tasks across multiple divergence settings and large language model backbones. These results demonstrate that GEB offers both a principled and practical solution for optimistic exploration in RLHF. Code is available here.

## 1 Introduction

Despite the acknowledged significance of online exploration for reinforcement learning with human feedback (RLHF) (Xu et al., 2024; Tang et al., 2024; Tajwar et al., 2024), there remains a paucity of theoretical frameworks governing *how to explore*. As shown in Fig. 1 (1, top), standard online RLHF algorithms (Guo et al., 2024; Pang et al., 2024; Ahmadian et al., 2024) generally rely on passive exploration, *i.e.*, the stochasticity of the policy itself to generate responses, with no mechanism to incentivize novelty or diversity. As a result, this approach can be notoriously sample-inefficient. When the optimal behavior resides in low-probability regions, passive exploration is unlikely to discover it, leading to policies that remain trapped around local optima.

To address this, some works (Das et al., 2024; Ji et al., 2024; Mehta et al., 2023; Qi et al., 2025; Liu et al., 2024) have attempted to devise sample-efficient algorithms, inspired by the principle *optimism in the face of uncertainty*. As illustrated in Fig. 1 (2, top), the principle aims to generate responses for regions of high epistemic uncertainty, thus encouraging data collection in unexplored areas for further training. To operationalize this, recent attempts (Zhang et al., 2024a; Xie et al., 2024; Cen et al., 2025) encourage exploration by adding *exploratory bonuses* to reward modeling, which is practically optimizeable for large language models. These methods intend to artificially inflate rewards in underexplored regions, nudging the policy toward more informative data collection.

Unfortunately, our theoretical analysis in Section 3 reveals a fundamental pitfall: under the common KL-regularized RLHF, the existing theoretical framework of exploratory bonuses fails to satisfy optimism. In particular, we prove that existing bonus formulations can undesirably drive the policy $\pi$ toward the reference policy $\pi_{\text{ref}}$ due to the divergence regulation in the exploratory bonus, and the induced bonus actually biases exploration toward high-probability regions of the reference model. As illustrated in Fig. 1 (II, bottom), the bonus disproportionately amplifies rewards for regions already well-covered by $\pi_{\text{ref}}$, thereby reinforcing conservative behavior rather than driving exploration into uncertain regions. This failure is not confined to KL-divergence; we further extend our analysis to the more general $\alpha$-divergence family and prove that the same collapse persists across a wide

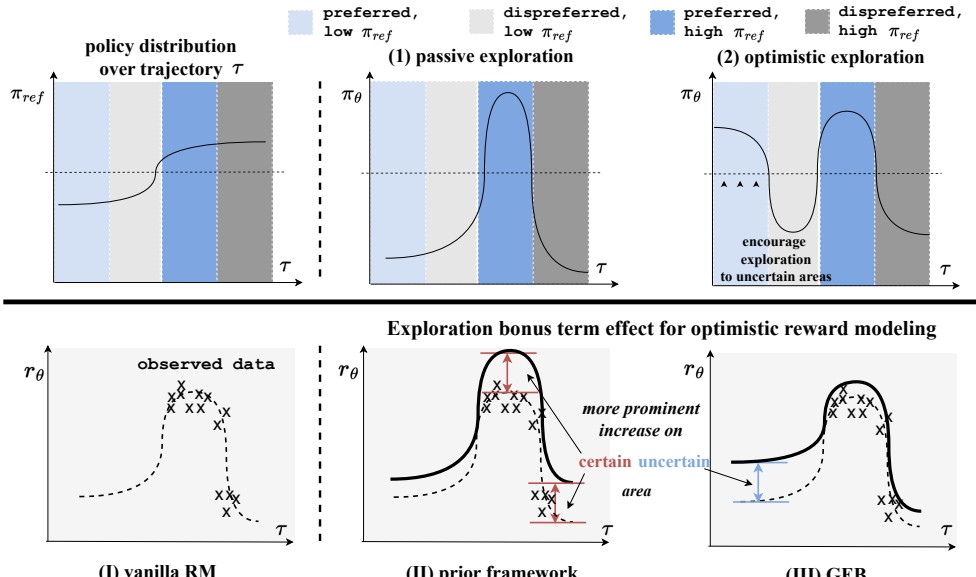

Figure 1: **The upper part** compares passive exploration and optimistic exploration. Optimistic exploration stimulates the trajectories $\tau$ of small $\pi_{ref}$ (seldom visited/uncertain), while passive exploration sticks to the high-$\pi_{ref}$ region, failing to approach global optima. The dashed line separates regions of high vs. low likelihood under the learning policy $\pi_\theta$. **The lower part** contrasts the effect of the exploration bonus term in optimistic reward modeling between prior works and our GEB. Prior works often emphasize rewards in frequently visited regions, which constrains exploration within certain areas. In contrast, our GEB amplifies rewards in seldom-visited regions, thereby encouraging further sampling in uncertain areas and successfully achieving optimistic exploration.

range of divergence-regularized objectives. Thus, while existing approaches appear to encourage exploration, they in fact undermine the very principle of optimism they aim to realize.

Motivated by these failures, we propose a new framework, **General Exploratory Bonus (GEB)**, which theoretically unifies existing approaches while provably satisfying optimism (Section 4). GEB corrects the failure modes of prior approaches by directly introducing a reference-dependent regulation into the reward. This adjustment offsets the undesired conservatism induced by divergence regularization, allowing the exploratory bonus to satisfy optimism—it increases the probability of responses rarely sampled to pursue potentially more preferred answers, as shown in Fig. 1 (III, bottom). Importantly, GEB provides a unified formulation: prior heuristic exploratory bonuses can be reinterpreted as special cases, and the framework naturally extends to the full class of $\alpha$-divergences. Beyond correcting the theoretical shortcomings, GEB remains practically implementable—it can be seamlessly integrated into the standard iterative RLHF loop without additional sampling cost.

We validate GEB on a large-scale alignment task across different divergences and model backbones. Empirically, GEB consistently yields stronger alignment compared to its counterpart of passive exploration. For example, the three GEB variants that we consider generally outperform the iterative f-DPO (Xiong et al., 2024a) across different divergence regulations, while the most performant variant surpasses several existing optimistic exploration methods that incorporate exploratory bonuses (Zhang et al., 2024a; Xie et al., 2024; Cen et al., 2025). By analyzing the distribution of sampled responses, we validate that GEB can successfully encourage sampling in the region of small $\pi_{ref}$, thereby effectively achieving optimistic exploration. We summarize our main contributions:

1. We formally prove that the existing theoretical framework of exploratory bonuses under KL and $\alpha$-divergence regularization fails to achieve optimistic exploration.

2. We introduce General Exploratory Bonus (GEB), a novel theoretical framework of optimistic exploration for RLHF that provably satisfies the optimism principle and unifies prior heuristic bonuses.

3. We empirically validate GEB on LLM alignment tasks, showing improved performance and broad applicability across multiple divergence families.

## 2    PRELIMINARIES

**Iterative online RLHF.** Let $x$ be a prompt sampled from a distribution $\rho$ and $y$ be a response given $x$, which is sampled from a policy $\pi(\cdot|x)$ modeled by a language model. We denote by $r(x, y)$ a real-valued reward model. An iterative online RLHF proceeds for rounds $T$, where each round $t = 1, ..., T$ has the following three steps: (i) The reward model $r_t$ is trained on the human preference dataset $\mathcal{D}_t = \{(x, y^w, y^l)\}$, where $y^w, y^l$ denote the preferred and dispreferred response to $x$; (ii) The policy $\pi_t$ is updated to maximize the reward $r_t(x, y)$ for responses $y \sim \pi_t(\cdot|x)$ conditioned on prompt $x$; and (iii) using the updated policy, we sample $\tilde{x} \sim \rho$, and generate multiple response pairs $(\tilde{y}_1, \tilde{y}_2) \sim \pi_t(\cdot|\tilde{x})$. Human evaluators then annotate these pairs to produce preference-labeled data $\{(\tilde{x}, \tilde{y}^w, \tilde{y}^l)\}$. The dataset for the next round is formed by $\mathcal{D}_{t+1} = \mathcal{D}_t \cup \{(\tilde{x}, \tilde{y}^w, \tilde{y}^l)\}$. For reward modeling step (i), we typically adopt the Bradley-Terry objective (Bradley & Terry, 1952):

$$r_t = \arg\min_r \mathcal{L}_{BT}(\mathcal{D}_t, r) = \arg\min_r \mathbb{E}_{(x,y^w,y^l)\sim\mathcal{D}_t} - \log[\sigma(r(x, y^w) - r(x, y^l))], \qquad (1)$$

where $\sigma$ denotes the sigmoid function. Next, in each step (ii), given the learned reward function $r_t$, the policy $\pi_t$ is updated to maximize the expected reward, often with a KL-regularization as follows

$$\pi_t = \arg\max_\pi \mathcal{J}_{\beta,\mathrm{KL}}(\pi, r_t) = \arg\max_\pi \mathbb{E}_{x\sim\rho, y\sim\pi(\cdot|x)} r_t(x, y) - \beta\mathbb{D}_{\mathrm{KL}}(\pi\|\pi_{\mathrm{ref}}), \qquad (2)$$

where $\beta > 0$ is a hyperparameter and $\pi_{\mathrm{ref}}$ is the reference model. The effectiveness of iterative online RLHF (Xiong et al., 2024b; Dong et al., 2024) has been validated in various real-world systems such as Claude (Bai et al., 2022) and LLaMA-series (Touvron et al., 2023; Grattafiori et al., 2024), but there is still much room for improvement in terms of sample-efficient exploration.

**Sample inefficiency of iterative online RLHF.** In online RLHF, standard online sampling is usually performed passively, relying solely on the LLM policy's inherent randomness. However, if the policy assigns a small probability to the optimal action, passive exploration may never explore it. Some recent theoretical analyses (Muldrew et al., 2024; Dong et al., 2024) and empirical evidence (Chen et al., 2025; Cen et al., 2025) present that the passive approach fails to sufficiently explore the prompt-response space. Particularly, Xie et al. (2024) demonstrates that the sample complexity can be exponential in $1/\beta$ for passive exploration, which is unacceptable in the small-$\beta$ regime. After then, follow-up studies propose to implement the principle "optimism towards uncertainty" into RLHF algorithms, i.e., encourage exploration of uncertain trajectories. Several works on this try to estimate uncertainty by leveraging some uncertainty quantification techniques, such as elliptical potential (Bai et al., 2022), Bayesian modeling (Qi et al., 2025), and epistemic neural network training (Liu et al., 2024). However, these methods are generally computationally prohibitive in LLM-scale settings. Therefore, recent works (Xie et al., 2024; Cen et al., 2025; Zhang et al., 2024a) propose *exploratory bonuses* for optimistic exploration, which can be computationally more tractable for LLM-based optimization.

## 3    EXPLORATORY BONUS AND HOW IT CAN FAIL

In this section, we first provide the iterative online RLHF formulation with an exploratory bonus (Section 3.1). We then theoretically prove that the existing formulation can fail to achieve optimistic exploration under both KL-constrained RLHF (Section 3.2) and a more general $\alpha$-divergence-regularized RLHF (Section 3.3), motivating our proposed method in Section 4.

### 3.1    EXPLORATORY BONUS

To improve the sample efficiency of iterative online RLHF, recent works (Zhang et al., 2024a; Cen et al., 2025) introduce exploratory bonuses, which aim to encourage the policy model to explore the under-visited space given an optimistic reward estimation. These approaches modify the standard RLHF loop by adding an exploratory bonus term $\mathcal{L}_{\mathrm{bonus}}$ in the reward modeling phase. Specifically, in the $t$-th iteration, the reward model $r_t$ and policy $\pi_t$ are optimized by

$$r_t = \arg\min_r \left[ \mathcal{L}_{BT}(\mathcal{D}_t, r) - \kappa\mathcal{L}_{\mathrm{bonus}}(r) \right], \qquad (3)$$

$$\pi_t = \arg\max_\pi \mathcal{J}_{\beta,\mathrm{KL}}(\pi, r_t) = \arg\max_\pi \mathbb{E}_{x\sim\rho, y\sim\pi(\cdot|x)} r_t(x, y) - \beta\mathbb{D}_{\mathrm{KL}}(\pi\|\pi_{\mathrm{ref}}), \qquad (4)$$

where $\kappa > 0$ is a hyperparameter. By Eq. 3, the reward model $r_t$ should not only fit the observed data in $\mathcal{D}_t$, but also learn to maximize the bonus term $\mathcal{L}_{\mathrm{bonus}}(r)$.

To boost exploration, the bonus term is designed to amplify the probability mass of policy more in underexplored areas rather than incentivizing it solely towards high empirical reward areas. As mentioned in § 2, early works on RLHF optimistic exploration are computationally prohibitive in the LLM fine-tuning regime. Thus, it is necessary to set a new approach that not only aligns with the principle of optimism in the face of uncertainty but is also cost-effective. For this, we derive a new condition for the exploration bonus to achieve optimism, avoiding direct uncertainty quantification:

**Definition 3.1 (Optimism condition for exploration bonus)** *Given an input prompt $x$ and a response $y$, when a reward model $r$ and a policy $\pi$ are computed with Eq. 3 and Eq. 4, respectively, the exploratory bonus $\mathcal{L}_{bonus}$ achieves optimism, if*

$$\frac{\partial}{\partial \pi_s(y|x)}\left(\frac{\partial \mathcal{L}_{bonus}(r(x,y))}{\partial \pi(y|x)}\right) < 0, \tag{5}$$

*where $\pi_s$ is a typical sampling policy, a joint policy on all iterations up to the current iteration.*

Specifically, at the $t$-th iteration, the typical sampling policy $\pi_s = \pi_1 \circ \pi_2 \circ \cdots \circ \pi_t$ is a joint distribution of all previous policies up to the current iteration. This distribution is not directly computable; rather, it serves as a theoretical construct describing how responses in $\mathcal{D}_t$ are generated. In Eq. 5, rather than characterizing $\mathcal{L}_{bonus}$ in its original function space, we define it by a condition on its partial derivatives with respect to two policies: current policy and typical policy. This new optimism condition not only enables us to flexibly define $\mathcal{L}_{bonus}$, but also serves as a core tool for theoretically analyzing existing methods. Although $\mathcal{L}_{bonus}$ appears unrelated to the current policy $\pi$—as it is defined in terms of the reward model $r(x,y)$—**the policy-reparameterized reward $r_\pi(x,y)$ allows us to express $r(x,y)$ directly in terms of** $\pi$ as follows (Rafailov et al., 2024):

$$r(x,y) := r_\pi(x,y) = \beta \log \frac{\pi(y|x)}{\pi_{\text{ref}}(y|x)} + \beta \log Z(x), \quad \text{where } Z(x) = \mathbb{E}_{y \sim \pi_{\text{ref}}} \exp(r(x,y)/\beta).$$

This is derived from the closed-form solution of the proximal preference optimization (Eq. 2) as $\pi(y|x) = \frac{\exp(r(x,y)/\beta)}{Z(x)}$. Thanks to this alternative representation of the reward, we can express and interpret the bonus term with respect to our current policy $\pi$, which yields the following implication.

> **Implication.** Our new optimism condition requires the derivative of the bonus term with respect to the current policy, $\partial \mathcal{L}_{\text{bonus}}(r_\pi(x,y))/\partial \pi(y|x)$, to be negatively correlated with the typical policy $\pi_s$. In other words, as a response $y$ is likely rare sample under $\pi_s$ (i.e., uncertain or underexplored responses), it should receive a larger ascending force in the policy distribution $\pi$, i.e., a higher $\partial \mathcal{L}_{\text{bonus}}(r_\pi(x,y))/\partial \pi(y|x)$. This new definition of the optimism principle, which is specified through the lens of partial derivative alignment, ensures the exploratory bonus nudges the exploration towards an uncertain response region without explicit uncertainty quantification. In practice, $\pi_s$ can be substituted by the reference model policy $\pi_{\text{ref}}$ or intermediate checkpoints of $\pi$ across iterations. We adopt the commonly used $\pi_{\text{ref}}$ as $\pi_s$ in our following demonstration.

## 3.2 Failure Under KL-constrained RLHF

Previous works, including Zhang et al. (2024a) and Cen et al. (2025), formulate the exploratory bonus with $\mathcal{L}_{\text{bonus}}(r) = \max_\pi \mathcal{J}_{\beta,KL}(\pi,r)$. Under this formulation, optimizing the exploratory bonus in Eq. 3 yields a min–max bilevel objective: $\min_r -\kappa \max_\pi [\mathbb{E}_{x \sim \rho, y \sim \pi} r(x,y) - \beta \mathbb{D}_{\text{KL}}(\pi \| \pi_{\text{ref}})]$. Intuitively, this objective encourages $r$ not only to fit the observed data via $\mathcal{L}_{BT}$ but also to assign high reward to unobserved regions by maximizing $\max_\pi \mathbb{E}_{x \sim \rho, y \sim \pi} r(x,y)$ in $\mathcal{L}_{\text{bonus}}(r)$. Here, we theoretically show that such formulations can suffer from optimism failures under KL-regularized RLHF.

**Lemma 3.1 (Optimism failure under KL-divergence.)** *Let $r_1 = \arg\min_r \mathcal{L}_{BT}(\mathcal{D}, r)$ be a reward model trained with the vanilla BT loss, and let $r_2 = \arg\min_r [\mathcal{L}_{BT}(\mathcal{D}, r) - \kappa \max_\pi \mathcal{J}_{\beta,KL}(\pi,r)]$ be a reward model trained with an additional exploratory bonus. If the policy is optimized via Eq. 4, then $r_1$ and $r_2$ yield the same set of policies.*

See the proof in Appendix B.1. The lemma shows that incorporating the exploratory bonus $\mathcal{L}_{\text{bonus}}(r) = \max_\pi \mathcal{J}_{\beta,\text{KL}}(\pi,r)$ into the reward training objective ***fails to induce the policy to sample***

*from low-$\pi_{ref}(y|x)$ regions, i.e., unexplored responses*. That is, $\mathcal{L}_{\text{bonus}}$ is ineffective for optimism. We next extend the result beyond KL divergence to a more general class of $\alpha$-divergence families.

### 3.3 GENERALIZATION TO $\alpha$-DIVERGENCE-CONSTRAINED RLHF

In this subsection, we theoretically show that the failure of optimism can broadly be extended to the $\alpha$-divergence class. Many common divergences, such as reverse KL-divergence, Hellinger distance, and forward KL-divergence, are special cases of $\alpha$-divergence.

Table 1: Realized exploratory bonus under different divergence classes when $\mathcal{L}_{\text{bonus}}(r) = \max_\pi \mathcal{J}_{\beta,f}(\pi, r)$.

| $f$ | exploratory bonus |
|---|---|
| reverse KL | constant |
| forward KL | $\mathbb{E}_{x\sim\rho, y\sim\pi_{\text{ref}}} \log \frac{\pi(y|x)}{\pi_{\text{ref}}(y|x)}$ |
| Hellinger distance | $\mathbb{E}_{x\sim\rho, y\sim\pi_{\text{ref}}} \sqrt{\frac{\pi(y|x)}{\pi_{\text{ref}}(y|x)}}$ |

**Definition 3.2 ($\alpha$-divergence class)** *An $\alpha$-divergence is a certain type of function $D(p|q) = \int f(\frac{\mathrm{d}p}{\mathrm{d}q})\mathrm{d}q$ that measures the difference between two probability distributions $p$ and $q$, where*

$$f(x) = \frac{x^\alpha - \alpha x - (1-\alpha)}{\alpha(1-\alpha)},$$

*and $\alpha$ is a hyperparameter typically with $0 \le \alpha \le 1$.*

**Lemma 3.2 (Optimism failure under $\alpha$-divergence.)** *Consider an objective $\mathcal{J}_{\beta,f}(\pi, r) = \mathbb{E}_{x\sim\rho, y\sim\pi(y|x)} r(x, y) + \beta \mathbb{E}_{x\sim\rho, y\sim\pi_{ref}(y|x)} f(\frac{\pi(y|x)}{\pi_{ref}(y|x)})$, where $f$ belongs to $\alpha$-divergence class. If a reward is trained with $\hat{r} = \arg\min_r[\mathcal{L}_{BT}(\mathcal{D}, r) - \kappa\mathcal{L}_{bonus}]$ and a policy $\pi$ is updated by $\arg\max_\pi \mathcal{J}_{\beta,f}(\pi, \hat{r})$ with $\mathcal{L}_{bonus} = \max_\pi \mathcal{J}_{\beta,f}(\pi, r)$, the gradient of the bonus satisfies $\frac{\partial^2 \mathcal{L}_{bonus}(r_\pi)}{\partial \pi_{ref}\partial\pi} \ge 0$, which means $\mathcal{L}_{bonus}$ encourage trajectories with large $\pi_{ref}$ more strongly, in contradiction to the optimism principle (Definition 3.1).*

*Proof* For a RL objective $\mathcal{J}_{\beta,f}(\pi, r)$, the relation between the optimal policy $\pi_f^*$ and the reward $r$ can be formulated as follows,

$$\pi_f^*(y|x) = \frac{1}{Z(x)}\pi_{\text{ref}}(y|x)(f')^{-1}(r(x,y)/\beta), \quad r_\pi(x,y) = \beta f'(\frac{\pi^*(y|x)}{\pi_{\text{ref}}(y|x)}Z(x)), \quad (6)$$

where $Z(x)$ is a normalization term and $(f')^{-1}$ is the inverse function of $f'$. The bi-level objective can be similarly transformed to a single level one by canceling the inner maximization $\max_\pi$ by Eq. 6. The single-level objective can be written as $r_t = \arg\min_r \mathcal{L}_{BT}(\mathcal{D}, r) - \kappa\mathbb{E}_{x\sim\rho, y\sim\pi_{\text{ref}}}\frac{1}{Z(x)}(f')^{-1}(\frac{r(x,y)}{\beta}) \cdot r(x,y) - \beta f(\frac{1}{Z(x)}(f')^{-1}(\frac{r(x,y)}{\beta}))$. Since the policy is computed by $\arg\max_\pi \mathcal{J}_{\beta,f}(\pi, r)$, the reward can be reparameterized by the policy with Eq. 6, which fortunately cancels $Z(x)$. Then, the optimistic reward-modeling objective can be reparameterized as

$$\arg\min_\pi \mathcal{L}_{dpo}(\mathcal{D}, \pi) - \kappa\beta\mathbb{E}_{x\sim\rho, y\sim\pi_{\text{ref}}}\left[\frac{\pi(y|x)}{\pi_{\text{ref}}(y|x)}f'(\frac{\pi(y|x)}{\pi_{\text{ref}}(y|x)}) - \beta f(\frac{\pi(y|x)}{\pi_{\text{ref}}(y|x)})\right]. \quad (7)$$

Since for $\alpha$-divergence, $f(u) = \frac{u^\alpha - \alpha u - (1-\alpha)}{\alpha(\alpha-1)}$, the partial derivative of Eq. 7 is $(\frac{\pi_{\text{ref}}}{\pi})^{1-\alpha}$, which induces positively correlated gradients w.r.t. $\pi$ and $\pi_{\text{ref}}$ when $0 \le \alpha < 1$, and is a constant when $\alpha = 1$, hence contradictory to the optimism defined in Definition 3.1. $\square$

According to Lemma 3.2, we summarize several forms of exploratory bonus induced by different $\alpha$-divergences in Table 1. For clarity, these expressions are presented after removing constant coefficients and additive biases. In every case, the resulting bonus encourages the policy to place more probability mass on responses that the reference model already samples frequently, rather than on underexplored responses. Now, we further prove that it actually drives $\pi$ to collapse toward $\pi_{\text{ref}}$ and that the failure extends beyond $\alpha$-divergence to other $f$-divergences.

**Theorem 3.3 (Optimism failure beyond $\alpha$-divergence.)** *When $f$ belongs to $f$-divergence, and the reward function is obtained by $\hat{r} = \arg\min_r[\mathcal{L}_{BT}(\mathcal{D}_t, r) - \kappa\max_\pi \mathcal{J}_{\beta,f}(\pi, r)]$ and the policy is updated by $\arg\max_\pi \mathcal{J}_{\beta,f}(\pi, \hat{r})$, the bonus term $-\kappa\max_\pi \mathcal{J}_{\beta,f}(\pi, r)$ induces the policy model $\pi$ to coincide with $\pi_{ref}$ when $xf''(x)$ is a monotone function.*

The monotonic increase of $xf''(x)$ can be satisfied by a broader divergence class beyond $\alpha$-divergence, including JS-divergence and Pearson $\chi^2$. Please see the detailed proofs in Appendix B.2.

> **Intuitive understanding.** The optimization of the exploratory bonus in Eq. 3 is a min-max bi-level objective, $\min_r -\kappa \max_\pi [\mathbb{E}_{x\sim\rho,y\sim\pi} r(x,y) - \beta \mathbb{D}_{\text{KL}}(\pi\|\pi_{\text{ref}})]$. Due to inner maximization $\max_\pi$, the divergence constraint implicitly makes $\pi$ close to $\pi_{\text{ref}}$ to reduce the KL divergence. Meanwhile, outer minimization $\min_r$ forces $r$ to provide high rewards in the region of high $\pi$ to maximize the expected reward. Their combination implicitly makes $r$ focus more on the region of high $\pi_{\text{ref}}$. As responses in high $\pi_{\text{ref}}$ area are easily sampled from scratch, prior exploratory bonuses just concentrate sampling on regions that are already frequently visited, *contradictory to the optimism principle, which requires encouraging exploration for responses $y$ rarely sampled by the reference model.*

## 4 GENERAL EXPLORATORY BONUS WITH OPTIMISM PRINCIPLE

Motivated by the failure of the existing optimistic exploration works, we now propose a novel framework, *General Exploratory Bonus* (**GEB**), and prove that it achieves optimism. We further show that prior heuristic bonuses—and their broader variants—emerge as special cases of our formulation.

**Formulation of a novel exploratory bonus.** As shown in the previous section, existing bonus schemes fail because the divergence constraints in $\max_\pi \mathcal{J}_{\beta,f}(\pi, r)$ force the optimal policy $\pi$ to remain close to $\pi_{\text{ref}}$, thereby biasing exploration toward regions where $\pi_{\text{ref}}$ is large. Achieving optimistic exploration requires the optimal $\pi$ to counteract this effect and deviate from $\pi_{\text{ref}}$. *Our key idea is therefore to incorporate an additional $\pi_{\text{ref}}$-dependent term into the reward that offsets the influence of the divergence regularization.* The resulting exploratory bonus takes the form

$$\mathcal{L}_{\text{bonus}} = \max_\pi J_{\beta,f}(\pi, R), \tag{8}$$

where our new reward formulation, $R$, now depends not only on the original reward model $r(x,y)$ but also on $\pi_{\text{ref}}(y|x)$. Now, the optimal policy of $\max_\pi J_{\beta,f}(\pi, R(x,y))$ can be obtained by replacing $r(x,y)$ with $R(x,y)$ in Eq 6. This yields $\pi^*(y|x) = \frac{1}{Z_R(x)}\pi_{\text{ref}}(f')^{-1}(\frac{R(x,y)}{\beta})$, where $Z_R(x)$ is a normalization term.

Following Lemma 3.2, we substitute $\pi(y|x)$ in $\max_\pi J_{\beta,f}(\pi, R(x,y))$ with its optimal form $\pi^*(y|x)$ and then apply the reward reparameterization trick for $\alpha$-divergences (Wang et al., 2024), i.e., $r(x,y) = f'(\pi^*(y|x)/\pi_{\text{ref}}(y|x))$, where $f$ specifies the divergence. Given this policy-reparameterized reward, we specify the exploratory bonus term in Eq 8 as follows:

$$\mathcal{L}_{\text{bonus}} = \beta \mathbb{E}_{x\sim\rho, y\sim\pi_{\text{ref}}(\cdot|x)}\left[\frac{u(x,y)}{Z_R(x)}f'(u(x,y)) - f(\frac{u(x,y)}{Z_R(x)})\right]. \tag{9}$$

In Eq. 9, we introduced $u(x,y)$ as an atomic function employed to construct the actual loss, and it is given by $u(x,y) = (f')^{-1}(R(x,y)/\beta)$ in this setup. Note that, after reward reparameterization, $u(x,y)$ can be expressed in terms of $\pi(y|x)$ and $\pi_{\text{ref}}(y|x)$. Moreover, as the functional form of $R(x,y)$ is not restricted to a specific class, $u(x,y)$ can be instantiated in many ways using these two distributions, while it must satisfy $u(x,y) > 0$ for all $x, y$ to ensure that the argument of $f'(\cdot)$ lies within its domain. See Table 2 for the example entries we are considering in this work.

**Equivalence to a practical objective.** In our proposed exploratory bonus, the normalization term $Z_R(x)$ in Eq. 9 cannot be eliminated. Fortunately, Lemma 4.1 (proved in Appendix B.4) shows that the training objectives with and without $Z_R(x)$ are equivalent. This equivalence allows us to convert the objective into a more concise form, facilitating both analysis and practical implementation.

**Lemma 4.1** *Denote two objectives as $h(u(x,y)) = \mathbb{E}_{x\sim\rho,y\sim\pi_{\text{ref}}}u(x,y)f'(u(x,y)) - f(u(x,y))$ and $\hat{h}(u(x,y)) = \mathbb{E}_{x\sim\rho,y\sim\pi_{\text{ref}}}\frac{u(x,y)}{Z_R(x)}f'(u) - f(\frac{u(x,y)}{Z_R(x)})$ where $u(x,y)$ is a function with $\pi(y|x)$ and $\pi_{\text{ref}}(y|x)$. If the ratio*

$$\frac{f'(u(x,y)) + u(x,y)f''(u(x,y)) - f'(\frac{u(x,y)}{Z_R(x)})}{Z_R(x)u(x,y)f''(u(x,y))} = \Lambda(x) \tag{10}$$

Table 2: GEB under different divergence classes and design of $u$. Note that (1) now the bonus term can be computed without the reference probability mass $\pi_{\text{ref}}$; (2) all of these $u$ instantiations meet the condition $u > \alpha$ when $0 < \pi < 1$. The presented bonuses are simplified by removing constants.

| $\mathcal{L}_{\text{bonus}}$ \\ $f$ | $1 + \alpha - \pi$ | $1/\pi$ | $\text{arctanh}(1 - \pi) + \alpha$ |
|---|---|---|---|
| reverse KL | $\mathbb{E}_{x\sim\rho,y\sim\pi_{\text{ref}}} - \pi(y\|x)$ | $\mathbb{E}_{x\sim\rho,y\sim\pi_{\text{ref}}} \frac{1}{\pi(y\|x)}$ | $\mathbb{E}_{x\sim\rho,y\sim\pi_{\text{ref}}}\text{arctanh}(1 - \pi(y\|x))$ |
| forward KL | $\mathbb{E}_{x\sim\rho,y\sim\pi_{\text{ref}}} \log(1 - \pi(y\|x))$ | $\mathbb{E}_{x\sim\rho,y\sim\pi_{\text{ref}}} - \log \pi(y\|x)$ | $\mathbb{E}_{x\sim\rho,y\sim\pi_{\text{ref}}} \log \text{arctanh}(1 - \pi(y\|x))$ |
| Hellinger Distance | $\mathbb{E}_{x\sim\rho,y\sim\pi_{\text{ref}}} \sqrt{1.5 - \pi(y\|x)}$ | $\mathbb{E}_{x\sim\rho,y\sim\pi_{\text{ref}}} \frac{1}{\sqrt{\pi(y\|x)}}$ | $\mathbb{E}_{x\sim\rho,y\sim\pi_{\text{ref}}} \sqrt{\text{arctanh}(1 - \pi(y\|x)) + 0.5}$ |

*is independent of $y$ and $\Lambda(x) > 0$, then minimizing the two objectives, $\min_\pi -h(u(x,y))$ and $\min_\pi -\hat{h}(u(x,y))$, yields the same class of optimal policies.*

Note that the $\alpha$-divergence ($0 \leq \alpha \leq 1$; see Def. 3.2) naturally satisfies the condition in Eq. 10 whenever $u(x,y) > \alpha$. As we show in the next paragraph, enforcing $u(x,y) > \alpha$ is straightforward in practice, which grants our framework substantial flexibility and extensibility. Leveraging Lemma 4.1, we can thus rewrite our objective in Eq. 9 into a concise and analytically convenient form without the normalization term:

$$\mathcal{L}_{\text{bonus}} = \beta\mathbb{E}_{x\sim\rho,y\sim\pi_{\text{ref}}}\Big[u(x,y)f'(u(x,y)) - f(u(x,y))\Big], \tag{11}$$

where $u(x,y)$ is flexibly formulated by $\pi(y|x)$ and $\pi_{\text{ref}}(y|x)$ satisfying $u(x,y) > \alpha$.

**GEB successfully achieves optimism.** Building on Lemma 4.1, we now show that our proposed framework achieves the optimism condition in Definition 3.1 (See Appendix B.5 for the proof).

**Theorem 4.2** *Consider an $\alpha$-divergence $f$ with $0 \leq \alpha \leq 1$, and the exploratory bonus $\mathcal{L}_{bonus} = \beta\mathbb{E}_{x\sim\rho,y\sim\pi_{ref}}\Big[u(x,y)f'(u(x,y)) - f(u(x,y))\Big]$, where $u(x,y)$ is a function dependent on $\pi(y|x)$ and $\pi_{ref}(y|x)$. For any $(x,y)$, if $\frac{\partial u}{\partial \pi} + \pi_{ref}\frac{\partial^2 u}{\partial \pi \partial \pi_{ref}} + \frac{(\alpha-1)\pi_{ref}}{u}\frac{\partial u}{\partial \pi}\frac{\partial u}{\partial \pi_{ref}} < 0$ and $u(x,y) > \alpha$, the optimism condition in Definition 3.1 is satisfied; that is, $\frac{\partial^2 \mathcal{L}_{bonus}}{\partial \pi \partial \pi_{ref}} \leq 0$.*

In our formulation, $u(x,y)$ can be flexibly defined in terms of $\pi(y|x)$ and $\pi_{\text{ref}}(y|x)$ as long as it satisfies the derivative condition in Theorem 4.2 and $u(x,y) > \alpha$. In particular, when $u(x,y)$ depends solely on $\pi(y|x)$ and is independent of $\pi_{\text{ref}}(y|x)$, any function that is strictly decreasing in $\pi$ with $u(x,y) > \alpha$ constitutes a valid choice. This design flexibility underscores the extensibility of our framework. In Table 2, we list several such choices of $u$, along with their corresponding reparameterized exploratory bonus terms under three different $\alpha$-divergences. From a practical standpoint, since $\mathcal{L}_{\text{bonus}}$ is computed as an expectation over $\pi_{\text{ref}}(\cdot|x)$, it does not require additional sampling and can be seamlessly integrated into iterative online RLHF. Meanwhile, to avoid unintended decreases in the likelihood of preferred responses, we follow Chen et al. (2025) and restrict the computation of the bonus on rejected responses to ensure that the probability of preferred responses continues to increase.

**Prior exploratory bonuses are encompassed within GEB.** Although we have shown that existing theoretical formulations of $\mathcal{L}_{\text{bonus}}$ fail to guarantee optimism, many practical implementations have nevertheless been effective through various approximations and adaptations. These approximations and adaptations are generally inextensible beyond the reverse KL divergence (detailed in Appendix B.3). In this paragraph, we show that these practical implementations can be naturally subsumed into our GEB framework, and even broader objectives can be reinterpreted as instances of optimistic exploration. For example, Zhang et al. (2024a) and Xie et al. (2024) finally implement their exploratory bonus as $\kappa\mathbb{E}_{x\sim\rho,y\sim\pi_{\text{ref}}(y|x)} \log \pi(y|x)$, which belongs to GEB when $u = -\log\pi + 1$ and $f$ is KL-divergence. Similarly, Cen et al. (2025) implement the exploratory bonus as $\kappa\mathbb{E}_{x\sim\rho,y\sim\pi_{\text{cal}}(\cdot|x)} \log \frac{\pi}{\pi_{\text{ref}}}$ where $\pi_{\text{cal}}$ is a fixed calibration distribution. This also falls under GEB by setting $u = -\frac{\pi_{\text{cal}}}{\pi_{\text{ref}}} \log \frac{\pi}{\pi_{\text{ref}}} - \frac{\pi_{\text{cal}}}{\pi_{\text{ref}}} \log \pi_{\text{ref}} + 1$ and $f$ is KL-divergence. Interestingly, even objectives not explicitly designed for exploration can be reinterpreted through our GEB framework. For instance, Chen et al. (2025) augment the DPO loss with an additional term

$\kappa \mathbb{E}_{x,y\sim\pi_{ref}}\sigma(-\beta\log\frac{\pi(y|x)}{\pi_{\text{ref}}(y|x)})$, which was originally introduced to control sample complexity. In our framework, this corresponds to optimistic exploration with $u = -\sigma(-\beta\log\frac{\pi(y|x)}{\pi_{\text{ref}}(y|x)}) + 1$.

> **Takeaways.** In summary, we introduce a general formulation of the exploratory bonus term, GEB, which showcases the following unique strengths: (1) In contrast to the prior bonus terms, GEB meets the optimism condition theoretically, thus provably boosts exploration on relatively untapped region; (2) GEB spans a broad class of instantiations as shown in Table 2, specified by the combination of the $\alpha$-divergence class and the functional form of $u$, which provides a plenty options for practitioners to choose on demand; (3) GEB is practical as it does not incur additional sampling costs and can be seamlessly integrated into the existing iterative online RLHF framework (Appendix C.1); and (4) GEB offers an unified understanding of existing methods by generalizing them as special cases in a single flexible formulation.

## 5 EXPERIMENTS

### 5.1 EXPERIMENTAL SETTINGS

Following prior works (Zhang et al., 2024a; Xie et al., 2024; Chen et al., 2025), we adopt the same iterative online algorithm as in Algorithm 1 with three iterations, aiming to isolate the effects of different exploration bonuses. We adopt two LLM backbones: Llama-3-8B-SFT (Dong et al., 2024) following prior works, and Mistral-Instruct-v0.3 (Jiang et al., 2023). The training prompt set is RLHFlow-UltraFeedback (Dong et al., 2024) as in previous works. URM-LLaMa-3.1-8B (Lou et al., 2024) serves as the preference oracle. We evaluate the outcome policies on both in-domain and out-of-domain test sets. Specifically, for the in-domain test, we use a held-out test set from UltraFeedback (Cui et al., 2024), and sample 64 times per prompt with the outcome policy to compare the average reward and win-rates against the base model. We use length-controlled AlpacaEval2 benchmark (Dubois et al., 2024) with GPT-4 as a judge for out-of-domain alignment test, and MATH-500 (Lightman et al., 2023) to evaluate out-of-domain reasoning ability.

**Baselines.** We adopt f-DPO (Wang et al., 2024), which extends DPO to the f-divergence class, as the primary baseline. We further compare GEB with three optimistic-exploration methods that incorporate exploratory bonuses—SELM (Zhang et al., 2024a), XPO (Xie et al., 2024), and VPO (Cen et al., 2025). Since the approximations or adaptations in their implementations do not extend beyond the KL divergence, we report their results only under KL. In contrast, we introduce a new baseline, Failed Exploratory Bonus (FEB), which removes these approximations or adaptations, i.e., Eq. 13.

### 5.2 RESULTS & ANALYSES

**GEB delivers robust improvements across different loss designs, divergence classes, and language model backbones.** The experimental results are shown in Table 3. Across both backbones, GEB generally outperforms f-DPO and FEB. Under the KL-divergence, GEB displays better or at least on-par performance compared to prior exploratory-bonus methods. Notably, the win-rate increases over 1.82% and 0.94% under the KL-divergence, over 2.36% and 1.29% under the Hellinger Distance, compared with their f-DPO counterpart. GPT-4 evaluation on the Alpaca benchmark also shows consistent performance gains on the out-of-domain alignment task. While GEB maintains on par, or usually better results in MATH, showing less performance degradation beyond alignment, known as alignment tax (Noukhovitch et al., 2023; Lin et al., 2024).

**GEB effectively encourages exploration in small $\pi_{\text{ref}}$ region, yielding more diverse sampling.** In Figure 2, we visualize the distribution of $\log \pi_{\text{ref}}$ for sampled responses in the last iteration under the KL divergence. When trained with the GEB, the policy model consistently samples more trajectories with a smaller $\pi_{\text{ref}}$ compared to the policy trained by f-DPO loss. This validates our motivation that GEB can encourage sampling trajectories of small $\pi_{\text{ref}}$ for optimistic exploration. In Table 5, we further calculate the distinct-n ($n = 1, 2, 3, 4$) for the sampled responses in the last iterations under the KL divergence, which measures the diversity of a corpus. GEB generally has higher diversity scores, validating that GEB incentivizes qualitatively more diverse samples.

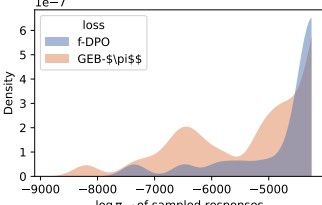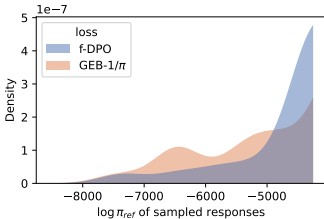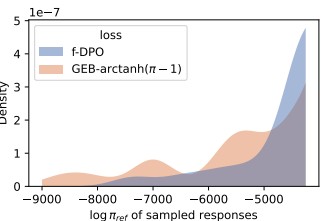

Figure 2: Comparison of $\log \pi_{\text{ref}}$ of sampled response in the last iteration between the general exploratory bonuses and vanilla iterative DPO. GEB-$\pi$, GEB-$1/\pi$, and GEB-$\text{arctanh}(\pi-1)$ corresponds to $1+\alpha-\pi$, $1/\pi$, and $\text{arctanh}(1-\pi)+\alpha$ as in Table 2

Table 3: In-domain evaluation on different exploration bonuses. **Boldface** and underline indicate the best and the second-best results, respectively. GEB-$\pi$, GEB-$1/\pi$, and GEB-$\text{arctanh}(\pi-1)$ corresponds to $1+\alpha-\pi$, $1/\pi$, and $\text{arctanh}(1-\pi)+\alpha$ as in Table 2.

| | KL ($\alpha$=1) | | Hel. ($\alpha$=0.5) | | f-KL ($\alpha$=0) | | Avg. | |
| --- | --- | --- | --- | --- | --- | --- | --- | --- |
| | WR | AvgR | WR | AvgR | WR | AvgR | WR | AvgR |
| *Mistral-Instruct-v0.3* | | | | | | | | |
| f-DPO | 78.42 | 0.7480 | 72.69 | 0.6536 | 51.11 | 0.5918 | 67.40 | 0.6645 |
| SELM | 77.56 | 0.7530 | - | - | - | - | - | - |
| XPO | 79.71 | 0.7492 | - | - | - | - | - | - |
| VPO | 78.57 | 0.7426 | - | - | - | - | - | - |
| FEB | 78.42 | 0.7480 | 71.54 | 0.6525 | 47.53 | 0.5928 | 65.83 | 0.6644 |
| GEB-$\pi$ | **81.00** | 0.7542 | 75.48 | **0.6641** | 51.68 | 0.5976 | **69.39** | 0.6720 |
| GEB-$1/\pi$ | 80.00 | 0.7554 | 73.97 | 0.6541 | 52.26 | **0.6051** | 68.74 | 0.6715 |
| GEB-$\text{arctanh}(\pi-1)$ | 79.71 | **0.7559** | **75.69** | 0.6614 | **52.76** | 0.5989 | **69.39** | **0.6721** |
| *LLaMA-3-8B-SFT* | | | | | | | | |
| f-DPO | 73.11 | 0.8050 | 71.11 | 0.7859 | 67.38 | 0.7579 | 70.53 | 0.7829 |
| SELM | 74.19 | 0.8126 | - | - | - | - | - | - |
| XPO | 72.40 | 0.8119 | - | - | - | - | - | - |
| VPO | 71.61 | 0.7971 | - | - | - | - | - | - |
| FEB | 73.11 | 0.8050 | 68.17 | 0.7591 | 67.95 | 0.7611 | 69.74 | 0.7751 |
| GEB-$\pi$ | 74.34 | **0.8156** | 71.68 | 0.7840 | 67.67 | **0.7681** | 71.23 | **0.7892** |
| GEB-$1/\pi$ | 74.76 | 0.8102 | 72.25 | 0.7859 | 68.17 | 0.7591 | 71.73 | 0.7851 |
| GEB-$\text{arctanh}(\pi-1)$ | **74.98** | 0.8080 | **73.26** | **0.7877** | **68.89** | 0.7569 | **72.38** | 0.7842 |

Table 4: Out-of-domain evaluation on different exploration bonuses with LLaMA-3-8B-SFT. **Boldface** and underline indicate the best and the second-best results, respectively. GEB-$\pi$, GEB-$1/\pi$, and GEB-$\text{arctanh}(\pi-1)$ corresponds to $1+\alpha-\pi$, $1/\pi$, and $\text{arctanh}(1-\pi)+\alpha$ as in Table 2.

| | KL($\alpha$=1) | | Hel.($\alpha$=0.5) | | f-KL($\alpha$=0) | | Avg. | |
| --- | --- | --- | --- | --- | --- | --- | --- | --- |
| | Alpaca | Math | Alpaca | Math | Alpaca | Math | Alpaca | Math |
| f-DPO | 25.72 | 67.6 | 24.73 | 69.0 | 17.80 | 69.2 | 22.75 | 68.6 |
| FEB | 25.72 | 67.6 | 23.75 | 68.6 | 19.62 | 68.6 | 23.03 | 68.3 |
| GEB-$\pi$ | **28.27** | 69.2 | 25.87 | 69.6 | **20.05** | **71.6** | **24.73** | **70.1** |
| GEB-$1/\pi$ | 26.10 | 68.4 | 25.28 | **70.2** | 19.80 | 69.2 | 23.73 | 69.3 |
| GEB-$\text{arctanh}(\pi-1)$ | 24.90 | **71.0** | **25.96** | 67.6 | 19.62 | 69.2 | 23.49 | 69.3 |

**The choice of $\kappa$.** Since the formulation of $u$ in Eq. 9 is flexible, the scale of the GEB term can differ substantially across designs, hence the absolute value of the bonus is less informative. Instead, we examine the relative ratio of the bonus term to the vanilla RL loss $|\kappa \mathcal{L}_{bonus}|/|\mathcal{L}_{RL}|$, which provides a more consistent basis for comparison and offers better practical guidance for tuning $\kappa$ across diverse settings. As shown in Fig. 3, performance remains stable when the ratio lies within a suitable range (1e-2 to 1e-6 in our case). However, if the ratio is too large, it impedes optimization of the RL objective and degrades performance; if too small, the exploration incentive in uncertain regions diminishes, and performance reverts to the vanilla baseline.

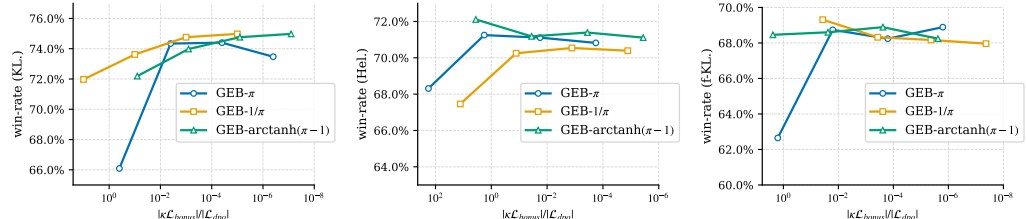

Figure 3: Experiments with different $\kappa$. The three graphs are under KL divergence, Hellinger Distance, and forward KL divergence from left to right, respectively. The p, f, tanh in the legends correspond to $1 + \alpha - \pi$, $1/\pi$, $\mathrm{arctanh}(1 - \pi) + \alpha$ in Table 2 respectively.

Table 5: Dist-n of the sampled corpus in the last iteration under the KL divergence.

|  | dist-1 | dist-2 | dist-3 | dist-4 |
|---|---|---|---|---|
| f-DPO | 0.0189 | 0.2700 | 0.6349 | 0.8418 |
| GEB-$\pi$ | 0.0192 | 0.2694 | 0.6323 | 0.8420 |
| GEB-$1/\pi$ | 0.0191 | 0.2738 | 0.6401 | 0.8448 |
| GEB-$\mathrm{arctanh}(\pi - 1)$ | 0.0192 | 0.2730 | 0.6391 | 0.8447 |

Table 6: The performance (Pass@16) of DPO and three GEB variants on math reasoning tasks.

|  | MATH500 | OlympiadBench | AIME 2025 | Avg. |
|---|---|---|---|---|
| DPO | 89.80 | 57.78 | 23.23 | 56.94 |
| GEB-$\pi$ | 92.80 | 64.59 | 28.23 | 61.87 |
| GEB-$1/\pi$ | 93.00 | 65.78 | 29.48 | 62.75 |
| GEB-$\mathrm{arctanh}\pi$ | 92.80 | 64.59 | 29.38 | 62.26 |

**GEB improves exploration in challenging problem instances.** To further illustrate when GEB offers the greatest benefit, we conduct an additional experiment targeting prompts whose preferred (i.e., correct) answers lie in regions rarely explored by the reference model during sampling. Starting from the DAPO-MATH-17K dataset (Yu et al., 2025), we filter out prompts for which the reference model, Qwen2.5-7B (Qwen et al., 2025), can generate a correct answer within two samples at temperature 0.6. This removes 5K prompts whose correct answers fall in frequently visited regions, leaving 12K prompts whose correct answers are less likely to be sampled. We use an open-source verifier, rather than reward models, to determine correctness. Each preference pair consists of one correct response and one wrong response. We set the learning rate to 5e-7 with a warm-up ratio of 0.1. For the reported runs under KL divergence, the $\kappa$ values for the three GEB variants are 50, 5e-2, and 5e-6, respectively.

As shown in Table 6, all three GEB variants consistently surpass the DPO baseline with a clear and stable performance margin. Notably, the performance gain of GEB is more than **6**% on Olympiad-Bench and AIME 2025. It indicates that our GEB can significantly enhance the exploration under scenarios where the correct answers lie in a region with a small $\pi_{\mathrm{ref}}$, i.e., an underexplored region. An additional experiment is provided in Appendix D.

## 6 CONCLUSION

While recent work proposes exploratory bonuses to operationalize the "optimism in the face of uncertainty" principle, our work shows that the existing theoretical frameworks of exploratory bonuses fail under KL and $\alpha$-divergence regularization. To address prior theoretical pitfalls, we introduce General Exploratory Bonus (GEB), a novel theoretical framework for sample-efficient RLHF. Our approach provably satisfies the optimism principle and unifies prior heuristic bonuses. We empirically validate GEB on LLM alignment tasks with diverse bonus designs and LLM backbones, showing improved performance and broad applicability across multiple divergence families.

## ACKNOWLEDGEMENT

We thank Froilan Choi and Shawn Im for their valuable suggestions on the paper, and thank all the ICLR 2026 PCs, SACs, ACs, and anonymous reviewers. This work is supported in part by the AFOSR Young Investigator Program under award number FA9550-23-1-0184, National Science Foundation under awards IIS-2237037 and IIS-2331669, Office of Naval Research under grant number N00014-23-1-2643, Schmidt Sciences Foundation, Open Philanthropy, Alfred P. Sloan Fellowship, and gifts from Google and Amazon.

## REPRODUCIBILITY STATEMENT

We have included all implementation details, hyperparameters, and training procedures in the paper and appendix. Our code and scripts for reproducing the experiments are publicly available through GitHub.

## ETHICS STATEMENT

This work studies reinforcement learning from human feedback (RLHF) using only publicly available or synthetic data, without new human subject collection. Here, by providing a rigorous theoretical framework with strong empirical evidence, we pursue a high standard of scientific excellence. We also take into account inclusiveness to make all our visualizations accessible to the unprivileged group of people by producing figures distinguished by light, shade, and marker. While RLHF has the potential to amplify biases or harmful behaviors if misused, our work is intended solely to advance safe and responsible research, and we encourage its application in alignment with ethical standards.

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

## A    RELATED WORKS

**Alignment & RLHF.**    Alignment (Hendrycks et al., 2021; Leike et al., 2018; Zhang et al., 2024b; 2025; Yeh et al., 2025) aims to ensure AI systems act in accordance with human values, preferences, and goals; and it has become a critical field in AI research. To steer language models to match human preferences, Reinforcement Learning from Human Feedback (RLHF) (Stiennon et al., 2020; Ouyang et al., 2022) achieves great success and has become the standard alignment pipeline. However, its computational complexity has motivated a family of Direct Preference Optimization (DPO) (Rafailov et al., 2024; Azar et al., 2024; Ethayarajh et al., 2024) that forgoes explicit reward modeling. Despite their efficiency, recent researchers (Tajwar et al., 2024; Xu et al., 2024; Xiong et al., 2024b) reemphasize the significance of online sampling.

**Optimistic exploration of RLHF.**    To address the computational overheads of passive exploration in RLHF, which samples trajectories just based on randomness, some existing attempts have been devoted to sample-efficient RL algorithms. Most of the works (Das et al., 2024; Ji et al., 2024; Dwaracherla et al., 2024; Mehta et al., 2023; Muldrew et al., 2024) adhere to the principle of optimism, proposing specialized prompt or response selection strategies to emphasize uncertain samples. While some research (Liu et al., 2024; Lou et al., 2024) propose uncertainty-aware reward models with epistemic neural networks or bootstrap ensembles, these methods introduce additional cost. Some research also addresses the sample efficiency with different theoretical foundations, such as information theory (Qi et al., 2025), preference-incentive exploration (Chen et al., 2025). Notably, several works (Zhang et al., 2024a; Xie et al., 2024; Cen et al., 2025) introduce different exploratory bonuses, which can implement optimism toward uncertainty without additional computations. However, they only focus on KL-divergence and their theoretical framework cannot result in real optimism as shown in Section 3.2.

**Efficient RL for LLM.**    Beyond optimistic exploration, some research proposes fine-grained signals for RL learning. For instance, several studies propose segment-level (Kong et al., 2025; Yin et al., 2025) or token-level (Zeng et al., 2024; Li et al., 2024) reward functions for alignment or text control. Notably, for reasoning tasks, the process reward model (Lightman et al., 2024; Li & Li, 2025; Yuan et al., 2024), which provides step-wise feedback for solutions, has shown promising effectiveness. On the other hand, recent research (Cheng et al., 2025; Zheng et al., 2025; Wu et al., 2025) on LLM reasoning reveals that high-entropy tokens guide the model toward diverse reasoning paths. Training with only high-entropy tokens is more beneficial for reasoning performance (Wang et al., 2025). While our approach is highly extensible, we believe the orthogonal methods can be further incorporated with our general exploratory bonus.

## B    OPTIMISM FAILURE OF PREVIOUS WORKS

### B.1    OPTIMISM FAILURE UNDER KL-DIVERGENCE

We start by proving how the existing exploratory bonus term under KL-divergence instantiation fails to achieve optimistic exploration.

**Lemma 3.1**    *Let $r_1 = \arg\min_r \mathcal{L}_{BT}(\mathcal{D}, r)$ be a reward model trained with the vanilla BT loss, and let $r_2 = \arg\min_r[\mathcal{L}_{BT}(\mathcal{D}, r) - \kappa \max_\pi \mathcal{J}_{\beta,KL}(\pi, r)]$ be a reward model trained with an additional exploratory bonus. If the policy is optimized via Eq. 4, then $r_1$ and $r_2$ yield the same set of policies.*

*Proof*    First, the inner maximization of the bonus term admits a closed-form solution, $\pi^*(y|x) = \pi_{\text{ref}}(y|x)e^{\frac{r(x,y)}{\beta}}/Z(x)$ where $Z(x) = \mathbb{E}_{y\sim\pi_{\text{ref}}(\cdot|x)}e^{\frac{r(x,y)}{\beta}}$ is a normalization term. Substituting this solution for the bi-level objective of $r_2$ reduces it to a single-level form:

$$r_2 = \arg\min_r \left[\mathcal{L}_{BT}(\mathcal{D}, r) - \kappa \mathbb{E}_{x\sim\rho}\beta \log \mathbb{E}_{y\sim\pi_{\text{ref}}} e^{\frac{r(x,y)}{\beta}}\right]. \tag{12}$$

As shown in Rafailov et al. (2024), the log-ratio $\beta \log \pi_\theta(y|x) - \beta \log \pi_{\text{ref}}(y|x)$ represents the same class of the original reward function $r$ through Eq. 4, thus all $r$ in the reward modeling objectives

can be reparameterized by the log-ratio. Plugging this into Eq. 12 yields

$$\arg\min_{\pi} \mathcal{L}_{dpo}(\mathcal{D}, \pi) - \kappa \mathbb{E}_{x\sim\rho} \beta \log \mathbb{E}_{y\sim\pi_{\text{ref}}(\cdot|x)} \frac{\pi(y|x)}{\pi_{\text{ref}}(y|x)}. \qquad (13)$$

Since the second term equals 0, the reparameterized Eq. 12 is exactly the vanilla DPO loss, that is, the reparameterized training objective of $r_1$. Thus, the exploratory bonus in the reward training objective does not affect the final policy set. □

## B.2 EXTENSION BEYOND $\alpha$-DIVERGENCE

The following theorem formally proves that the exploratory bonus $-\kappa \max_{\pi} \mathcal{J}_{\beta,f}(\pi, r)$ cannot encourage optimism for a more general divergence class.

**Theorem 3.3** *When $f$ belongs to $f$-divergence, the reward function is obtained via $\hat{r} = \arg\min_r[\mathcal{L}_{BT}(\mathcal{D}_t, r) - \kappa \max_{\pi} \mathcal{J}_{\beta,f}(\pi, r)]$, and the policy is updated by $\arg\max_{\pi} \mathcal{J}_{\beta,f}(\pi, \hat{r})$, the bonus term $-\kappa \max_{\pi} \mathcal{J}_{\beta,f}(\pi, r)$ induces the policy model $\pi$ to coincide with $\pi_{ref}$ when $xf''(x)$ is a monotone function.*

*Proof* By Lemma 3.2, we can reparameterize the bonus term used for optimistic reward modeling into Eq. 6. Denote $h(u) = uf'(u) - f(u)$. For a fixed prompt $x$, the training step can then be written as the following constrained optimization problem:

$$\arg\max_{\pi} \mathbb{E}_{y\sim\pi_{\text{ref}}(\cdot|x)} h(\frac{\pi(y|x)}{\pi_{\text{ref}}(y|x)}) \quad s.t. \quad \sum_y \pi(y|x) = 1 \quad \text{and} \quad \forall y, \pi(y|x) > 0. \qquad (14)$$

Then we can apply the Lagrange multiplier as

$$\mathcal{L} = \mathbb{E}_{y\sim\pi_{\text{ref}}} h(\frac{\pi(y|x)}{\pi_{\text{ref}}(y|x)}) - \mu\Big(\sum_y \pi(y|x) - 1\Big) - \sum_y \eta(y)\pi(y|x), \qquad (15)$$

where $\mu, \eta$ are the dual variables. Then we utilize the Karush-Kuhn-Tucker (KKT) conditions for the given optimization problem. The complementary slackness gives that $\forall y, \eta(y)\pi(y|x) = 0$. The stationary condition requires

$$\frac{\partial \mathcal{L}}{\partial \pi(y|x)} = h'(\frac{\pi(y|x)}{\pi_{\text{ref}}(y|x)}) - \mu - \eta(y) = 0. \qquad (16)$$

Let $S_y = \{y|\pi(y|x) > 0\}$. For all $y \in S_y$, complementary slackness implies $\eta(y) = 0$. Since $h'(u) = uf''(u)$ is a monotone function, we can obtain $\forall y \in S_y, \frac{\pi(y|x)}{\pi_{\text{ref}}(y|x)}$ is a constant. Then applying the normalization constraint, $\mathbb{E}_{y\sim\pi_{\text{ref}}(\cdot|x)} \frac{\pi(y|x)}{\pi_{\text{ref}}(y|x)} = 1$. Hence, we conclude that the unique interior optimum is $\pi^*(y|x) = \pi_{\text{ref}}(y|x)$. □

The theorem implies that the reparameterized exploratory bonus attains its maximum only when $\pi$ and $\pi_{\text{ref}}$ coincide. The condition that $xf''(x)$ is a monotone function is satisfied by $\alpha$-divergence and beyond, e.g. Pearson $\chi^2$. Hence, the exploratory bonus $-\kappa \max_{\pi} \mathcal{J}_{\beta,f}(\pi, r)$ in the reward training objective generally contradicts the optimism, since it cannot encourage trajectories with small initialized possibility.

## B.3 PRIOR ADAPTIONS & APPROXIMATIONS CANNOT GENERALIZE

Although the theoretical justification for the previously proposed exploratory bonus does not hold, its empirically implemented loss remains effective due to various adaptations and approximations. In this subsection, however, we show that these adaptations and approximations cannot be extended beyond the KL-divergence class.

Zhang et al. (2024a) modify the formulation of $\mathcal{J}_{\beta,f}(\pi, r)$ in the optimistic exploratory bonus $-\kappa \max_{\pi} \mathcal{J}_{\beta,f}(\pi, r)$ as

$$\mathcal{J}'_{\beta,f}(\pi, r) = E_{x,y\sim\pi,y'\sim\pi_{\text{ref}}}[r(x, y) - r(x, y')] - \beta D_{KL}(\pi|\pi_{\text{ref}}), \qquad (17)$$

which adds a bias in the reward expectation term. Under KL-divergence, the original $\mathcal{J}_{\beta,f}(\pi, r)$ will be zero after re-parameterization as shown in Lemma 3.1, thus the sole reparameterized bias term

will remain as $-\mathbb{E}_{y' \sim \pi_{\text{ref}}} \log \pi(y'|x)$. Since $\mathcal{J}_{\beta,f}(\pi, r)$ cannot be reparameterized to zero except KL-divergence, this adaptation cannot generalize then.

In the derivations of Cen et al. (2025), an idealized calibration distribution $\pi_{cal}$ is assumed to satisfy $\mathbb{E}_{y \sim \pi_{cal}} r(x, y) = 0$. Since $\pi_{cal}$ is not directly accessible in practice, the method substitutes rejected responses to approximate expectations under $\mathbb{E}_{\pi_{cal}}$. These rejected samples, however, do not satisfy the defining property of $\pi_{cal}$, making the approximation theoretically inconsistent.

The theoretical framework of Xie et al. (2024) is based on Implicit Q-Approximation (refer to Lemma C.3 in the original paper) as follows,

$$\beta \frac{\log \pi(y|x)}{\log \pi_{\text{ref}}(y|x)} = r(x, y) - V^*(x) \tag{18}$$

where $V(x)^*$ is the KL-regularized value function. Because this relation depends critically on the logarithmic structure of the KL divergence, the framework cannot be extended to more general divergence families either.

In contrast, our general exploratory bonus integrates seamlessly with iterative online RLHF algorithms and extends naturally to the entire $\alpha$-divergence family. As discussed in §4, all these heuristic bonus terms can be encompassed by our unified theoretical framework.

## B.4 EQUIVALENCE BETWEEN A SOPHISTICATED OBJECTIVE AND A SIMPLE ONE.

**Lemma 4.1** *Denote two objectives as $h(u(x,y)) = \mathbb{E}_{x \sim \rho, y \sim \pi_{ref}} u(x,y) f'(u(x,y)) - f(u(x,y))$ and $\hat{h}(u(x,y)) = \mathbb{E}_{x \sim \rho, y \sim \pi_{ref}} \frac{u(x,y)}{Z(x)} f'(u) - f(\frac{u(x,y)}{Z(x)})$ where $u(x,y)$ is a function with $\pi(y|x)$ and $\pi_{ref}(y|x)$. If the ratio*

$$\frac{f'(u(x,y)) + u(x,y) f''(u(x,y)) - f'(\frac{u(x,y)}{Z(x)})}{Z(x) u(x,y) f''(u(x,y))} = \Lambda(x) \tag{19}$$

*is independent of $y$ and $\Lambda(x) > 0$, then minimizing the two objectives, $\min_\pi -h(u(x,y))$ and $\min_\pi -\hat{h}(u(x,y))$, yields the same class of optimal policies.*

*Proof* Similar to Lemma 4.1, for a fixed $x$, we can write a similar formulation of the Lagrange multipliers as follows,

$$\mathcal{L} = \mathbb{E}_{y \sim \pi_{\text{ref}}} h(u(x,y)) - \mu_1 \left( \sum_y \pi(y|x) - 1 \right) - \sum_y \eta_1 \pi(y|x), \tag{20}$$

where $\mu, \eta$ are the dual variables. Then, we utilize the KKT conditions formulation for the given optimization problem. Similar to Lemma 4.1, when $\pi(x, y) > 0$, we obtain

$$\frac{\partial h}{\partial \pi(y|x)} = u(x,y) f''(u(x,y)) \cdot \pi_{\text{ref}}(y|x) \cdot \frac{\partial u(x,y)}{\partial \pi(y|x)} = \mu_1. \tag{21}$$

Similarly, we can obtain the KKT conditions for $\hat{h}(u(x,y))$ as follows,

$$\frac{\partial \hat{h}}{\partial \pi(y|x)} = \frac{1}{Z(x)} \left( f'(u(x,y)) + u(x,y) f''(u(x,y)) - f'(\frac{u(x,y)}{Z(x)}) \right) \cdot \pi_{\text{ref}}(y|x) \cdot \frac{\partial u(x,y)}{\partial \pi(y|x)} = \mu_2, \tag{22}$$

where $\mu_2$ is another dual variable. With the condition of Eq. 20, these two partial derivatives in Eq. 21 and Eq. 22 are equivalent when $\mu_2 = \mu_1 \Lambda(x)$. Hence, every policy that satisfies the stationary condition for $h$ also satisfies it for $\hat{h}$. Since $\Lambda(x) > 0$, the second-order derivative $\frac{\partial^2 h}{\partial^2 \pi(y|x)}$ and $\frac{\partial^2 \hat{h}}{\partial^2 \pi(y|x)}$ have the same sign, which indicates they share the same local minima. Hence, minimizing the two objectives $\min_\pi -h(u)$ and $\min_\pi -\hat{h}(u)$ induces the same class of policies.

$\square$

## B.5 GEB ENABLES OPTIMISTIC EXPLORATION

In this subsection, we prove how GEB meets the optimism principle specified by Definition 3.1.

**Theorem 4.2** *Consider an $\alpha$-divergence $f$ with $0 \leq \alpha \leq 1$, and the exploratory bonus $\mathcal{L}_{bonus} = \beta\mathbb{E}_{x \sim \rho, y \sim \pi_{ref}}\Big[u(x,y)f'(u(x,y)) - f(u(x,y))\Big]$, where $u(x,y)$ is a function dependent on $\pi(y|x)$ and $\pi_{ref}(y|x)$. For any $(x,y)$, if $\frac{\partial u}{\partial \pi} + \pi_{ref}\frac{\partial^2 u}{\partial \pi \partial \pi_{ref}} + \frac{(\alpha-1)\pi_{ref}}{u}\frac{\partial u}{\partial \pi}\frac{\partial u}{\partial \pi_{ref}} < 0$ and $u(x,y) > \alpha$, the optimism condition in Definition 3.1 is satisfied; that is, $\frac{\partial^2 \mathcal{L}_{bonus}}{\partial \pi \partial \pi_{ref}} \leq 0$.*

*Proof* First, substituting the optimal solution of the inner maximization and utilizing reward reparameterization, we obtain the training objective as in Eq. 9. By Lemma 4.1, this can be equivalently expressed as

$$\mathcal{L}_{\text{bonus}} = \beta\mathbb{E}_{x \sim \rho, y \sim \pi_{\text{ref}}(\cdot|x)}\Big[u(x,y)f'(u(x,y)) - f(u(x,y))\Big]. \tag{23}$$

For $\alpha$-divergences, the conditions in Lemma 4.1 are satisfied. Since we have $f'(u) + uf''(u) - f'(\frac{u}{Z}) = u^{\alpha-1}(Z^{1-\alpha} - \alpha)/(1-\alpha)$ and $uf''(u) = u^{\alpha-1}$, the fraction $\Lambda(x) = (Z^{1-\alpha} - \alpha)/Z(1-\alpha)$ in Lemma 4.1 is independent of $y$. Since $u > \alpha$, $Z_R = \mathbb{E}_{y \sim \pi_{\text{ref}}(\cdot|x)}u > 0$, thus $\Lambda(x) > 0$ is also satisfied. Finally, the mixed second-order derivative of Eq. 11 is computed as

$$\frac{\partial^2 \mathcal{L}_{\text{bonus}}}{\partial \pi \partial \pi_{\text{ref}}} = \beta\mathbb{E}_{x \sim \rho}\sum_y u^{\alpha-1}\Big(\frac{\partial u}{\partial \pi} + \pi_{\text{ref}}\frac{\partial^2 u}{\partial \pi \partial \pi_{\text{ref}}} + \frac{(\alpha-1)\pi_{\text{ref}}}{u}\frac{\partial u}{\partial \pi}\frac{\partial u}{\partial \pi_{\text{ref}}}\Big) < 0, \tag{24}$$

which achieves the optimism defined in Definition 3.1. $\qquad\qquad\square$

## C  EXPERIMENTS

### C.1  IMPLEMENTATION DETAILS

**Algorithm.** Following prior work (Zhang et al., 2024a; Xie et al., 2024; Chen et al., 2025), we adopt the same algorithmic backbone for empirical validation in order to isolate and compare the effects of different exploratory bonuses in the loss function. This backbone bypasses reward modeling at each iteration through reward reparameterization, a procedure commonly referred to as iterative DPO (Dong et al., 2024). Previous methods of exploratory bonuses further reparameterize their bonus terms to make them compatible with this framework. To extend the original framework to $\alpha$-divergence, we extend the iterative DPO with f-DPO (Wang et al., 2024) loss, and likewise reparameterize our generalized exploratory bonus accordingly. The full procedure is summarized in Algorithm 1. Since the only difference is the loss function in Line 5 of Algorithm 1, our GEB does not induce any additional compute costs.

---

**Algorithm 1** Iterative Online Algorithm with Exploratory Bonus

**Input:** Reference model $\pi_{\text{ref}}$, iteration number $T$, prompt set for each iteration $\mathcal{D}_1, \ldots, \mathcal{D}_T$, reward function $r$;
**Output:** Trained model $\pi_T$;
1: **for** iteration t = 1, 2, . . . , T **do**
2:     **for** $x \in \mathcal{D}_t$ **do**
3:         $y_1, y_2 \sim \pi_{\text{ref}}(\cdot|x)$ and obtain the rewards $r(y_1), r(y_2)$;
4:         Rank the reward and denote $y^+, y^-$ as the preferred and dispreferred response between $y_1, y_2$ and update $D_t = \{x, y^w, y^l\}$;
5:         $\pi_t = \arg\min_\pi \mathcal{L}_{\text{DPO}} - \kappa\mathcal{L}_{bonus}(\pi)$
6:         update $\pi_{\text{ref}}$ with $\pi_t$ (optional)
7:     **end for**
8: **end for**

---

**Hyperparameter settings and environments.** All experiments are conducted on two NVIDIA H200 GPUs. When training and sampling, the max length is set to 2048. For training, the batch size per device is set to 2; we enable gradient checkpointing, and the gradient accumulation step is set to 64; the learning rate is 5e-7 with a cosine scheduler, and the warm-up ratio is 0.03. In the main experiments, we use the best performance with $\kappa$ with a suitable ratio range to f-dpo loss across $1, 1e-2, 1e-4, 1e-6, 1e-8$. For sampling, the temperature is set to 1. For in-domain evaluation and MATH evaluation, we set temperature to 0.6 and top-p to 0.9; we use the default setting of alpaca-eval.

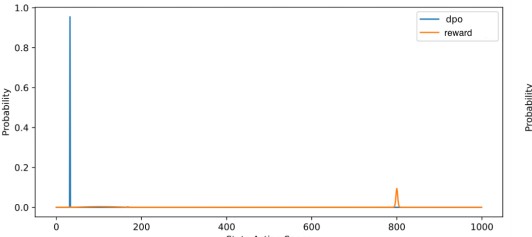 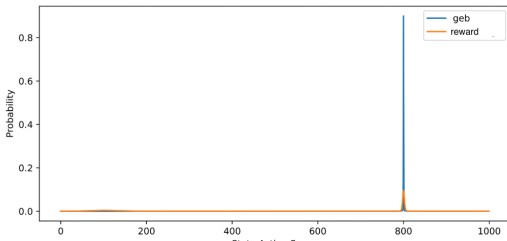

Figure 4: Comparison on the bandit policy distributions trained with DPO (left) and GEB (right). The DPO policy collapses to a local optimum, while the GEB policy continues to explore and ultimately chooses the globally preferred action.

## D  TOY EXPERIMENTS

To illustrate a setting in which GEB yields substantial improvement, we construct a toy example in which the most preferred action lies in a rarely visited region.

### D.1  EXPERIMENTAL SETTING

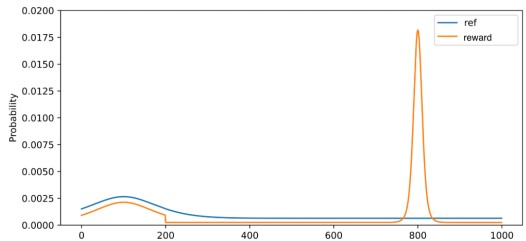

Figure 5: Initial reference bandit distribution ("ref") and the reward distribution. Because the most preferred action lies in a low-probability region, it is rarely visited under purely passive exploration.

We consider a 1000-arm bandit with 1000 parameters, each parameter corresponding to a distinct arm. As shown in Fig. 5, the most preferred action lies in a rarely visited region, making it unlikely to be sampled under pure passive exploration, as in f-DPO. Each experiment is run for 5000 iterations. At each iteration, the bandit policy generates 64 rollouts to form a batch of 32 preference pairs. The learning rate is set to 1e-2 with no warm-up phase.

### D.2  RESULTS AND ANALYSES

As shown in Fig. 4, the bandit policy trained with DPO becomes trapped in a local optimum: its probability mass collapses onto a suboptimal action because the policy never encounters the truly preferred action during training. In contrast, all three GEB variants successfully recover the desired distribution (right panel of Fig. 4), concentrating probability on the most preferred action. This demonstrates that GEB effectively promotes exploration into low-probability regions, enabling the policy to discover and select the optimal action despite its small initial likelihood.

## E  SUPPLEMENT RESULTS OF MAIN EXPERIMENTS

### E.1  REPEATED RUN OF THE MAIN EXPERIMENT

To assess the statistical significance of GEB's performance gains, we repeat each experiment five times and report the mean and standard deviation. As shown in Table 7, GEB achieves consistently higher performance than baseline methods, with statistically significant improvements. Moreover, we conduct independent p-test between GEB variants and f-DPO. The p-value is shown in Table 8. Most p-values are less then 0.05, which means the improvement of GEB is statistically significant.

### E.2  SEMANTIC COHERENCE OF THE SAMPLED RESPONSES

We use GPT-4 to evaluate whether a given sentence is coherent, nonsensical, or contains meaningless content. The scoring scale ranges from 0 (fully coherent) to 3 (complete nonsense with no coherent meaning). We apply this evaluation to the responses generated in the final training iteration of DPO and the three GEB variants. The resulting scores are as follows.

Table 7: Repeated in-domain evaluation on different exploration bonuses. **Boldface** and underline indicate the best and the second-best results, respectively. GEB-$\pi$, GEB-$1/\pi$, and GEB-$\mathrm{arctanh}(\pi-1)$ corresponds to $1 + \alpha - \pi$, $1/\pi$, and $\mathrm{arctanh}(1-\pi) + \alpha$ as in Table 2.

| | KL ($\alpha=1$) | | Hel. ($\alpha=0.5$) | | f-KL ($\alpha=0$) | |
|---|---|---|---|---|---|---|
| | WR | AvgR | WR | AvgR | WR | AvgR |
| *Mistral-Instruct-v0.3* | | | | | | |
| f-DPO | 77.49 ±1.32 | 0.7426 ±0.0201 | 72.70 ±0.96 | 0.6488±0.0114 | 50.83±0.62 | 0.5902±0.0330 |
| GEB-$\pi$ | **80.60**±0.96 | 0.7531±0.0162 | 74.41±0.72 | **0.6723**±0.0211 | 51.75±0.51 | 0.5986±0.0134 |
| GEB-$1/\pi$ | 79.69±1.10 | **0.7614**±0.0048 | 73.87±0.48 | 0.6342±0.0032 | **51.86**±0.60 | **0.6088**±0.0025 |
| GEB-$\mathrm{arctanh}(\pi-1)$ | 79.60±1.01 | 0.7544±0.0242 | **74.57**±0.95 | 0.6642±0.0231 | 51.69±1.03 | 0.5962±0.0153 |
| *LLaMA-3-8B-SFT* | | | | | | |
| f-DPO | 72.89 ±0.86 | 0.7926±0.0320 | 71.19±0.68 | 0.7915±0.0101 | 66.62±0.96 | 0.7566±0.0049 |
| GEB-$\pi$ | 73.90±1.10 | 0.8048±0.0086 | 72.08±0.74 | **0.7921** ±0.0031 | 67.79±1.14 | **0.7698**±0.0036 |
| GEB-$1/\pi$ | 74.36±0.92 | **0.8084**±0.0043 | 72.28±0.78 | 0.7733±0.0143 | 67.99 ±0.75 | 0.7618±0.0021 |
| GEB-$\mathrm{arctanh}(\pi-1)$ | **74.55**±1.02 | 0.8032±0.0019 | **72.57**±1.03 | 0.7902±0.0035 | **68.34**±0.76 | 0.7612±0.0130 |

Table 8: p-value of independent p-test between win-rates of GEB variants and f-DPO

| f-DPO vs. | Mistral-Instruct-v0.3 | | | LLaMA-3-8B-SFT | | |
|---|---|---|---|---|---|---|
| | KL. | Hel. | f-KL. | KL. | Hel. | f-KL. |
| GEB-$\pi$ | 0.003 | 0.014 | 0.034 | 0.145 | 0.084 | 0.117 |
| GEB-$1/\pi$ | 0.021 | 0.051 | 0.027 | 0.031 | 0.047 | 0.037 |
| GEB-$\mathrm{arctanh}(\pi-1)$ | 0.023 | 0.015 | 0.155 | 0.024 | 0.042 | 0.014 |

Table 9: Semantic coherence scores of responses produced by policy models trained using DPO and the GEB variants.

| | DPO | GEB-$\pi$ | GEB-$1/\pi$ | GEB-$\mathrm{arctanh}\pi$ |
|---|---|---|---|---|
| KL. | 1.24 | 1.08 | 1.32 | 1.19 |
| Hel. | 1.33 | 1.42 | 1.12 | 1.23 |
| fKL. | 1.32 | 1.38 | 1.32 | 1.30 |

Table 10: Ablation study on the sample responses used for the bonus term calculation. We compare the results on UltraFeedback (win rate) across three divergences with all responses (all) and with the rejected responses only (rejected-only).

| Method | KL all | KL rejected-only | Hel. all | Hel. rejected-only | fKL all | fKL rejected-only |
|---|---|---|---|---|---|---|
| GEB-$\pi$ | 79.78 | **81.00** | 73.40 | **75.48** | 51.32 | **51.68** |
| GEB-$\frac{1}{\pi}$ | 79.56 | **80.00** | 72.25 | **73.97** | 51.61 | **52.26** |
| GEB-$\mathrm{arctan}\pi$ | 79.64 | **79.71** | 73.11 | **75.69** | 51.25 | **52.76** |

As shown in Table 9, the responses produced by the GEB variants exhibit semantic coherence comparable to those generated by DPO. This indicates that, in practice, GEB promotes exploration into moderately underrepresented yet still semantically meaningful regions of the output space.

## E.3 THE CHOICE OF $u$

The three variants of the exploration bonuses in Table 2 represent different instantiations of the GEB framework. Each satisfies the optimism condition in Definition 3.1 and exhibits consistent performance improvements on the alignment task.

Nonetheless, the curvature of $u$ with respect to $\pi$ meaningfully affects the optimization dynamics of the exploratory bonus. For instance, when $u = 1 - \pi + \alpha$, the function is linear, thus the gradient of $u$ to $\pi$ is a constant. Therefore, the per-trajectory incentive to decrease $\pi$ is constant. In contrast, when $u$ is convex—such as $u = 1/\pi$—the gradient magnitude diminishes as $\pi$ becomes larger. This results in a more conservative reduction of $\pi$ compared to the linear case.

Takeaway: The curvature of $u$ with respect to $\pi$ governs the behavior of the exploration bonus. Greater convexity leads to a more conservative shift of probability mass toward underexplored regions.

**Restricting the exploratory bonus to rejected responses**    Restricting the bonus term to rejected responses is a common practice in prior works on exploratory bonuses (Zhang et al., 2024a; Chen et al., 2025; Cen et al., 2025). Importantly, this restriction does not constitute a theoretical departure from our framework. The optimism guarantee in Theorem 4.2 hinges on increasing the probability of low $\pi_{\text{ref}}$ regions, i.e., underexplored regions. Because rejected samples lie precisely in these low-probability areas, applying the bonus only to rejected responses preserves the intended optimism direction. While preserving the theoretical guarantee on the optimism, it also shows a practical advantage as shown in Table 10.

