# OpenReview forum: "General Exploratory Bonus for Optimistic Exploration in RLHF"
_ICLR.cc/2026/Conference — ICLR 2026 Poster_

### Official Review · Reviewer_e552 · 2025-10-24

**Soundness:** 3
**Presentation:** 2
**Contribution:** 2
**Rating:** 4
**Confidence:** 2

**Summary:**

The paper addresses the issue of exploration in LLM finetuning with RLHF. The authors show that a previously proposed novelty bonus does not induce more exploration and extend this insight from the proposed KL-divergence to the class of $\alpha$-divergences. Finally the authors define a condition they claim to be optimistic and develop a general exploratory bonus (GEB) based on it. Evaluation against other exploratory bonuses appears promising.

**Strengths:**

The addressed problem of insufficient exploration during LLM fine tuning is relevant and the analysis of the existing exploratory bonus is insightful.

**Weaknesses:**

I need to clarify that I am not an LLM expert, and am not familiar with any of the cited literature. As such I had huge problems following the paper's story and math. While I would reject this paper due to my lack of comprehension, I am willing to follow the recommendation of reviewers more experienced in this field. I admit that some of my problems might be trivial for experts in the field, but the fact that an outsider cannot follow some of the notations and definitions should not be ignored either.

In more detail:
- While the intuition in Figure 1 is very clear, I was wondering whether optimistic exploration (as used in RL literature [1]) is even desirable in LLM fine tuning. Most of the possible trajectory space is not sampled by the reference model and exploring those parts seems counter-productive: most of those trajectories will not follow proper grammar and are out-of-distribution for the learned RLHF reward function (which might therefore prefer them although they are garbage). Both are reasons to stay away from trajectories that have very small probabilities under the reference policy. So which trajectories do the authors *actually* propose to explore?
- Even after repeated attempts, I was unable to make sense of Definition 3.1. In RL literature there is a clear definition of optimism for exploration [1], which requires to estimate an epistemic upper bound on the true return/reward of an action. How is Def.3.1 related to that? What is the intuition behind an "ideal sampling distribution" $\pi_s$? Later the authors mention they "adopt the commonly use $\pi_{ref}$ as $\pi_s$", but I fail to see what the 2nd order derivative of the exploration bonus w.r.t. the current policy $\pi$ (that's what it is, right?) and the reference policy $\pi_{ref}$ has to with optimism.
- The introduced formalism has many undefined symbols or at least does not explain some default symbols. For example, I do not understand what the "policy-reparameterized reward model" in line 167 is. The left-hand side is a function of a policy, but the right hand side contains queries $x$ and completions $y$. Where do these come from? In line 163 it is defined as $r(x,y)=r(\pi)$, so are $x$ and $y$ just omitted in the function signature? What is the idea behind this reward and what does it have to do with Def.3.1? Another example is the appearance of $f'$ and $f''$ in the proof of Lemma 3.2: is this a partial derivative? If so which one? Why did the authors not use the clearer $\frac{\partial f}{\partial x}$ formalism they relied on earlier?
- Theorem 4.2 appears grammatically malformed. The first sentence contains an if, but no then part. Later there is a $\forall (x,y)$ quantifier, but no $x$ or $y$ appear in the theorem. The meaning of these things might be obvious to someone familiar with the literature, but a mathematical statement must be self-contained.
- The experiments do not contain standard deviation and I could not find out whether they have been repeated at all. This might be common in LLM literature, where repeating experiments can be prohibitively expensive, but I was unable to gauge whether improving from 79.71 to 81.00 is a worthwhile improvement or just stochastic noise.


[1] Lattimore T, Szepesvari C. Bandit Algorithms. Cambridge University Press; 2020.

**Questions:**

See above.

---

> ### Author Response · Authors · 2025-11-22
> **Response to Reviewer e552 (1/3)**
>
> We thank the reviewer for the thoughtful comments and suggestions. Below, we respond to your comments in detail.
>
> > W1 Whether optimistic exploration is desirable in LLM fine-tuning. Which trajectories do the authors actually propose to explore?
>
> Thank you for the insightful question! The need for optimistic exploration in LLM scenarios has been demonstrated and validated in several prior studies [1][2].
>
> While low-probability trajectories under the reference model may include semantically invalid outputs, these regions _also contain potentially uncertain yet high-reward behaviors_ that the current policy has never encountered. Without explicitly sampling them, the model cannot learn to distinguish genuinely poor responses from unobserved but promising ones. Optimistic exploration thus serves to ensure that the policy learns which outputs are truly undesirable.
>
> Empirically, GEB _does not push the policy toward arbitrarily low-probability trajectories_ (i.e., garbage), but rather to **encourage exploration within moderately underrepresented yet semantically meaningful regions** that the reference model underexplores. As shown in the table below, we use GPT-4 to score whether a given sentence is nonsensical, garbage, or coherently written. The score range is 0 (perfectly coherent) to 3 (total nonsense: no coherent meaning at all). We score the responses in the last iteration for three GEB variants, and the scores are as follows.
>
> | |DPO|GEB-$\pi$|GEB-$1/\pi$|GEB-$\text{arctanh} \pi$
> |---|---|---|---|---|
> |KL.| 1.24 | 1.08 | 1.32 | 1.19 |
> |Hel.| 1.33 | 1.42 | 1.12 | 1.23 |
> |f-KL.|1.32 | 1.38 | 1.32 | 1.30|
>
>
> From the table, the responses of GEB variants have on-par semantic coherence compared to those of the DPO baseline. It shows that GEB **encourages exploration within moderately underrepresented yet semantically meaningful regions** in practice. The results and the discussion have been supplemented in `Appendix F.2`.
>
> [1] Tengyang Xie, Dylan J. Foster, Akshay Krishnamurthy, Corby Rosset, Ahmed Awadallah, and Alexander Rakhlin. Exploratory preference optimization: Harnessing implicit q*-approximation for sample-efficient RLHF. ICLR 2025.
>
> [2] Wei Xiong, Hanze Dong, Chenlu Ye, Ziqi Wang, Han Zhong, Heng Ji, Nan Jiang, Tong Zhang. Iterative Preference Learning from Human Feedback: Bridging Theory and Practice for RLHF under KL-constraint. ICML 2025.

---

> ### Author Response · Authors · 2025-11-22
> **Response to Reviewer e552 (2/3)**
>
> > W2 How is Def. 3.1 related to the definition of optimism for exploration in RL literature?
>
> That's another insightful question. Our notion of optimism in Definition 3.1 draws inspiration from the RL literature, but it is adapted to the LM policy optimization setting. In classical RL, optimism is implemented by constructing an upper confidence bound on the (unknown) return, as you correctly pointed out. However, as demonstrated in `Line 136-143`, estimating an epistemic upper bound often leads to expensive calculation of the covariance matrix, which is computationally prohibitive in LLM-scale settings. In contrast, adding an exploration bonus (rewarding under-explored responses) is a computationally efficient alternative that implicitly encourages optimistic exploration, i.e., encourages sampling in seldom-visited regions.
>
> The Definition 3.1 formalizes the **sufficient condition** for a bonus term to realize optimistic exploration, i.e., encourage sampling in seldom-visited regions. Intuitively, _the "ideal sampling distribution" is the joint distribution of all previous policies up to the current iterations._ For example, in the $t$-th iteration, $\pi_s=\pi_1\circ\pi_2\circ\cdots\circ\pi_t$. This distribution is not directly computable; rather, it serves as a theoretical distribution from which we sample all existing sampling responses. Equation (5) expresses that the exploration bonus should up-weight trajectories with small sampling probability under $\pi_s$, thereby promoting exploration in less frequently sampled regions. To make the $\pi_s$ more intuitive, we renamed it from _"ideal sampling distribution"_ to _**"typical sampling policy"**_ in our revised draft (Definition 3.1).
>
> The first-order derivative $\frac{\partial\mathcal{L}{\text{bonus}}(r)}{\partial \pi(y|x)}$ represents the gradient that indicates how minimizing the bonus term encourages $\pi(y|x)$. The second-order derivative $\frac{\partial^2 \mathcal{L}{\text{bonus}}(r)}{\partial \pi(y|x)\partial \pi_s(y|x)}$ measures how this encouragement varies with respect to $\pi_s$. We require this second-order derivative to be negative, ensuring that the bonus strength increases as the sampling probability decreases, i.e., larger encouragement toward less $\pi_s$. **This property precisely matches the desired “optimism” behavior** illustrated in Figure 1 (III). In contrast, as demonstrated in Theorem 3.3, prior theoretical frameworks have _positive_ second-order derivatives, which means they tend to reinforce already well-explored regions (Fig. 1 (II)).
>
> We have revised our manuscript to clarify these points further. We add further clarification of $\pi_s$ in `Line 176-179`, and expand the implications of Definition 3.1 in `Line 190-200`.

---

> ### Author Response · Authors · 2025-11-22
> **Response to Reviewer e552 (3/3)**
>
> > W3-1 The introduced formalism has many undefined symbols, or at least does not explain some default symbols. What is the "policy-reparameterized reward model" in line 167?. What is the idea behind this reward, and what does it have to do with Definition 3.1?
>
> Thank you for pointing that out! For better clarity, we further introduce notations in `Line 110-118`, carefully revised the notation of reward reparameterization from $r(x,y)=r(\pi)$ to $r(x,y)=r_{\pi}(x,y)$, and elaborate on why we introduce this reparameterization in `line 181-200` of the revised manuscript.
>
> As mentioned in the above response, we introduced a new sufficient condition for optimistic exploratory bonus in Definition 3.1, which allows us to investigate the bonus term $L_{bonus}(r)$ whether it meets optimism or not through its partial derivatives with respect to the policy $\pi$. However, as the basic form of the bonus term is solely defined over the reward model, it is hard to interpret what the derivative of $r$ w.r.t. $\pi$ means. _This is why we need to introduce the reparameterization of reward_, $r(x,y)=r_{\pi}(x,y)=\beta \log\frac{\pi(y|x)}{\pi_{\text{ref}(y|x)}}+\log Z(x)$, in `line 184-185`. **This new reward formulation, which we called policy-reparameterized reward model, expresses the reward function with the current policy $\pi$ and the reference policy $\pi_{\text{ref}}$, enabling us to interpret the optimism condition for the bonus term (Def 3.1) through the lens of its partial derivative alignment.**
>
> After the DPO paper was introduced [1], this policy-reparameterized reward expression has become **a widely used technique** in the RLHF literature [2],[3] for interpreting and implementing the implicit reward model solely through the policy model.
>
> We believe the revised version has improved for broader accessibility -- thanks again for your helpful feedback.
>
> [1] Rafael Rafailov, Archit Sharma, Eric Mitchell, Christopher D Manning, Stefano Ermon, and Chelsea Finn. Direct preference optimization: Your language model is secretly a reward model. Advances in Neural Information Processing Systems, 2024.
>
> [2] Chaoqi Wang, Yibo Jiang, Chenghao Yang, Han Liu, Yuxin Chen. Beyond Reverse KL: Generalizing Direct Preference Optimization with Diverse Divergence Constraints. ICLR 2024.
>
> [3] Han Zhong, Zikang Shan, Guhao Feng, Wei Xiong, Xinle Cheng, Li Zhao, Di He, Jiang Bian, Liwei Wang. DPO Meets PPO: Reinforced Token Optimization for RLHF. ICML 2025.
>
> > W3-2 Clarification on $f'$ and $f''$ in the proof of Lemma 3.2
>
> The $f'$ and $f''$ in Lemma 3.2 are not partial derivatives. Since the definition of $f$ in Definition 3.2 has only **one variable**, $f'(\cdot)$, $f''(\cdot)$ denote the first and second **ordinary derivatives** of $f$ with respect to its single variable.
>
> > W4 Theorem 4.2 appears grammatically malformed.
>
> Thanks for the careful suggestion! We have revised it and uploaded the new version.
>
> > W5 Standard deviation in experiments
>
> To demonstrate that our performance gain is significant rather than a luck of hyperparameter or noise, we run three repetitions for DPO and three for each GEB variant, and report their average win rate and standard deviation. The results are provided in the table below, and are also supplemented in our revised draft, `Appendix F.1`. **The results show that GEB's performance gain is statistically significant**.
>
> Additionally, GEB’s advantages would be most pronounced when optimal behaviors reside in low-$\pi_{\mathrm{ref}}$ regions—i.e., areas underexplored by the reference policy. To demonstrate the problem instances, we include **two additional experiments**.
>
> The first experiment is a toy example of a 1000-arm bandit, where the most preferred action is designed to lie in a region with low reference probability, which means it is unlikely to be sampled during exploration. The results show that the final policy trained with DPO is trapped in local optima, while the policy trained by GEB variants can successfully discover the most preferred action and achieve the global optimum (see `Figure 4`).
>
> The second experiment is on a challenging math reasoning task. We filter out the easy prompts that can be correctly answered by the reference model within two trials. The filter ensures that the correct answers of the remaining prompts lie in a region with less probability, which are more difficult to sample out during exploration. The experiments show that for the Pass@16 performance on three math benchmarks, the model trained by GEB variants can outperform the one trained by DPO with a **substantial performance gap** (**+7%** on Olympiad Bench).
>
> |Method|MATH500 | Olympiad Bench |AIME 2025| Average|
> |---|---|---|---|---|
> |DPO|89.80| 57.78|23.23 |56.94 |
> |GEB-$\pi$| 92.80| 64.59| 28.23 | 61.87  |
> |GEB-$1/\pi$ | 93.00| 65.78|29.48 |  **62.75**  |
> |GEB-$\text{arctanh}\pi$|92.80| 64.59| 29.38 | 62.26 |
>
> We have added these two additional experiments in `Appendix E` and `Line 431-482`.

---

> ### Comment · Reviewer_e552 · 2025-11-24
> **Thanks to the authors for their rebuttal.**
>
> I am still unsure about your answers about your answer to W1, though:
> - Your answer did not address my concern about out-of-distribution rewards functions. Any optimistic (upper bounded) exploration that is allowed to go into regions not seen during training will find wrongly generalized samples with extremely high reward. What keeps your method from converging to those? Is it the iterative nature that will label these regions in the next iteration and therefore prevent the policy to converge to nonsensical predictions $y$?
> - You claim that GEB "encourages exploration within moderately underrepresented yet semantically meaningful regions". But why? What keeps the exploration away from *strongly underrepresented* or *semantically meaningless* regions? Is such a statement even possible without a tight definition of "semantically meaningful"? In the additional results the "nonsensicallity" (of one can trust GPT4 here) seem to fluctuate quite a bit with the used metrics.

---

> > ### Comment · Reviewer_e552 · 2025-11-24
> >
> > - I appreciate the clarification in W3-1 how the "policy-reparameterized reward" is derived. There are a could minor errors: l.188: Solving (eq.2) analytically yields the solution $\pi(y|x) = \frac{1}{Z(x)} \pi_{ref}(y|x) \exp(r(x, y)/\beta)$. Solving this for $r(x,y)$ yields second summand $+ \beta \log Z(x)$, where the $\beta$ is missing in l185-186.
> >
> > - I also appreciate the additional runs for W5. However, almost all of the new results in Table 7 have overlapping standard deviations with f-DPO, and you can not claim "The results show that GEB's performance gain is statistically significant" unless you perform more rigorous statistical significance tests!
> >
> > - The additional math reasoning test are very nice, but, again, without repetitions it is hard to say whether this is a significant effect or just noise.

---

> > > ### Comment · Reviewer_e552 · 2025-11-24
> > > **Still not convinced about the definition of optimism**
> > >
> > > Lastly, the added description of your "optimism condition" is welcome, but I still fail to see whether this is a meaningful insight. I understand now what you mean with "typical sampling distribution", and I see that $\frac{\partial L_{bonus}(x, y)}{\partial\pi_{ref}(y|x)} < 0$ incentivizes bonuses that decrease where the visitation probability increases. However, I do not see why another derivative of that w.r.t. $\pi(y|x)$ is supposed to constitute a trade-off with the reward. I understand the policy-reparametrized reward now (see above), but I fail to see how it fits into this discussion: the derived $r(x,y)$ only refers to the real and bonus rewards of the previous iteration.
> > >
> > > I think what I miss is a clear formulation of the exploration-exploitation tradeoff, something like r_{real}(x,y) + \alpha r_{bonus}(x,y). A intrinsic reward formulation like this induces an upper bound (or at least an upper bound on an upper bound, see O'Donoghue et al, 2018), which in RL would be called "optimism". Without such a clear link I find it hard to accept Definition 3.1 as a "sufficient condition for a bonus term to realize optimistic exploration". I apologize if I misunderstand this, but your claim sounds almost circular: we define something as a sufficient condition, and it is a sufficient condition because we defined it like that".
> > >
> > > Could you please try to formulate your argument as: (i) here is a generally accepted form of optimism and (ii) here is a theorem showing that for all optimistic bonuses holds this property? This would make it much easier for RL experts like me to accept that the following conclusions have actual merit.
> > >
> > >
> > > **References**
> > >
> > > O'Donoghue et al., 2018: https://arxiv.org/abs/1709.05380

---

> ### Author Response · Authors · 2025-12-03
> **Further response to Reviewer e552 (1/2)**
>
> Thanks for your time reading our response. We are happy to clarify your questions further.
>
> > How does the exploration bonus that allows sampling in the underexplored region and finding samples with extremely high reward?
>
> Thank you for the insightful question. Exactly as you mentioned in your question: It is due to the iterative nature. After all the newly generated responses are labeled in the next iteration, the policy can learn from not converging to nonsensical predictions.
>
>
> > What keeps the exploration away from strongly underrepresented or semantically meaningless regions? In the additional results, the "nonsensicality" seems to fluctuate quite a bit with the used metrics.
>
> Our experiment in Appendix F.2 provides an empirical clue that the sampling responses of GEB achieve similar semantic coherence compared to GEB. Theoretically, it would originate in the combination of the encouragement of the GEB exploratory bonus in Eq.3 and the f-divergence regulation of policy objectives in Eq.4.
>
> Moreover, whether our exploratory bonus can be theoretically proved to encourage sampling in "semantically meaningful" would not affect our theoretical argument.
>
> - First, due to the iterative nature, the nonsensical responses would naturally become dispreferred responses in the next iteration. As we mentioned in our initial response, while low-probability trajectories under the reference model may include semantically invalid outputs, these regions also contain potentially uncertain yet high-reward behaviors that the current policy has never encountered. Without explicitly sampling them, the model cannot learn to distinguish genuinely poor responses from unobserved but promising ones. Optimistic exploration thus serves to ensure that the policy learns which outputs are truly undesirable.
>
> - Our paper aims to reveal that previous theoretical frameworks of exploratory bonuses fail to achieve optimistic exploration, while GEB can. Whether the exploration is semantically meaningful would not affect the process of optimistic exploration.
>
>
> > $\beta$ in policy-reparameterized reward.
>
> Thanks for your suggestion. We have revised the formulation accordingly.
>
>
> > Statistical significance tests.
>
> Thank you for the suggestion. We followed your advice and reran all experiments with 5 runs. We then performed a t-test comparing three GEB variants with f-DPO. As shown in Table 8, p-values are generally less than 0.05, which suggests that the performance improvement is significant.
>
>
> > I see that $\frac{\partial L_{bonus}}{\partial \pi_{ref}}$ incentivizes bonuses that decrease where the visitation probability increases. Why is another derivative of that w.r.t. $\pi$ supposed to constitute a trade-off with the reward?
>
> Thank you for the insightful question. We have revised the second-order partial derivatives in Eq.5 to clarify the order. **$\frac{\partial L_{bonus}}{\partial \pi_{ref}}$ is not the first-order derivative in our definition. It alone cannot convey any practical meaning here, since $\pi_{ref}$ is not even a trainable variable.**
>
> As we mentioned in Line 191-120 and the previous response, **the first-order derivative is $\frac{\partial\mathcal{L}_{\text{bonus}}(r)}{\partial \pi(y|x)}$**, which represents how minimizing the bonus term encourages $\pi(y|x)$. The second-order derivative $\frac{\partial^2 \mathcal{L}{\text{bonus}}(r)}{\partial \pi(y|x) \partial \pi_s(y|x)}$ measures how this encouragement varies with respect to $\pi_s$. We require this second-order derivative to be negative, ensuring that the bonus strength increases as the sampling probability decreases, i.e., larger encouragement toward less $\pi_s$. **This property precisely matches the desired “optimism” behavior** illustrated in Figure 1 (III). In contrast, as demonstrated in Theorem 3.3, prior theoretical frameworks have _positive_ second-order derivatives, which means they tend to reinforce already well-explored regions (Fig. 1 (II)).

---

> ### Author Response · Authors · 2025-12-03
> **Further response to Reviewer e552 (2/2)**
>
> > Build a connection to the traditional optimism
>
> (i) A generally accepted form of optimism
>
> We take the form in your question $r_{real}(x,y) + \alpha r_{bonus}(x,y)$ as a generally accepted form of optimism. For optimistic exploration, $r_{bonus}$ decreases where the visitation increases. $\frac{\Delta r_{bonus}}{\Delta \pi_s}<0$.
>
>
> (ii) connection to Definition 3.1
>
> Since $r_{t+1} = r_t - \eta (\frac{\partial L_{BT}}{\partial r} - \kappa \frac{\partial L_{bonus}}{\partial r} )$ where $\eta$ is learning rate, we can regard $$\Delta r_{bonus} \propto \frac{\partial L_{bonus}}{\partial r}, \Delta r_{real} \propto \frac{\partial L_{BT}}{\partial r}.$$ Since $\frac{\Delta r_{bonus}}{\Delta \pi_s}<0$, we have $$\frac{\partial}{\partial \pi_s}(\frac{\partial L_{bonus}}{\partial r}) <0.$$
>
> With the reward reparameterization in Line 185-186, we have $\frac{\partial L_{bonus}(r_\pi)}{\partial \pi}=\frac{\partial L_{bonus}(r)}{\partial r}\frac{\partial r}{\partial \pi}$. Since $\frac{\partial r_\pi}{\partial \pi} = \frac{\beta}{\pi}$, we have $$ \frac{\beta}{\pi} \frac{\partial}{\partial \pi_s}(\frac{\partial L_{bonus}}{\partial \pi}) <0. $$
> Since $\frac{\beta}{\pi}>0$, we have $$  \frac{\partial}{\partial \pi_s}(\frac{\partial L_{bonus}}{\partial \pi}) <0, $$
> which is exactly our Definition 3.1.

---

### Official Review · Reviewer_3ha8 · 2025-10-31

**Soundness:** 1
**Presentation:** 1
**Contribution:** 1
**Rating:** 2
**Confidence:** 3

**Summary:**

This paper aims to address limitations in existing exploratory bonus formulations for reinforcement learning from human feedback (RLHF). The authors claim that prior approaches under KL or α-divergence regularization fail to achieve true “optimism in the face of uncertainty.” They propose the General Exploratory Bonus (GEB) framework, which theoretically corrects these failures and empirically improves performance on several RLHF benchmarks.

While the paper tackles an important question, it suffers from serious clarity, organization, and presentation issues. The writing quality is poor, making the technical content extremely difficult to follow. Furthermore, the claimed theoretical and empirical contributions are not clearly or convincingly demonstrated.

**Strengths:**

The topic (exploration in RLHF) is timely and relevant. The attempt to provide a unified theoretical treatment across divergence families could be valuable if clearly presented.

**Weaknesses:**

The paper’s language is riddled with grammatical errors and awkward phrasing. Many sentences are either ungrammatical or semantically unclear, which severely hinders readability. Several paragraphs are almost unreadable due to convoluted sentence structures and a lack of coherent logical flow.
While Figure 1 conveys an important idea, its design is confusing: it is unclear why a line connects the four sections instead of simply displaying their values. This creates the impression of a continuous transition between them, which undermines the intended clarity.

**Questions:**

Your optimism condition (Definition 3.1 and Theorem 4.2) is central to the paper, but its mathematical and conceptual justification remains unclear. Could you clarify why it is an appropriate formalization of the “optimism in the face of uncertainty” principle, and how it connects to standard exploration metrics (e.g., uncertainty, entropy, or epistemic variance)?

Most reported improvements (e.g., in Tables 3 and 4) appear to be within 1–2 % over baselines. Could you provide more evidence that these differences are statistically significant and not due to noise or hyperparameter sensitivity?

---

> ### Author Response · Authors · 2025-11-22
> **Response to Reviewer 3ha8 (1/2)**
>
> We thank the reviewer for the comments and suggestions. Below, we respond to your comments and questions in detail.
>
> > W1 Writing clarity
>
> We appreciate the reviewer’s feedback regarding the clarity and readability of the manuscript. Following your comments, **we have undertaken a substantial revision of the writing throughout the entire paper**, with particular emphasis on the Method section (Please see our revised draft!). By deferring the theoretical proofs to the Appendix, we added much more clarification and intuition to the main paper. We believe these revisions significantly improve the presentation.
>
> > W2 Clarification on Figure 1
>
> The line in Figure 1 was not meant to suggest a continuous transition between the four cases but rather to **separate regions of high vs. low likelihood under the policy**. The figure illustrates how different exploration schemes allocate probability mass—from passive exploration (favoring high-$\pi_\text{ref}$ regions) to optimistic exploration (encouraging low-$\pi_\text{ref}$ regions).
>
> We have revised the `caption of Figure 1` to better clarify this.
>
>
> > Q1 Why is Definition 3.1 an appropriate formalization of the “optimism in the face of uncertainty” principle, and how does it connect to standard exploration metrics (e.g., uncertainty, entropy, or epistemic variance)?
>
> Thank you for the insightful question.
>
> **Connection to the classical optimism and standard exploration metrics, such as uncertainty?**: Our notion of optimism in Definition 3.1 draws inspiration from the optimism in the face of the uncertainty principle, but it is adapted to the LM policy optimization setting. In classical RL, optimism is implemented by constructing an upper confidence bound on the (unknown) return. However, as mentioned in `Line 136-141`, estimating an epistemic upper bound often leads to computationally expensive calculation of the covariance matrix, which is computationally prohibitive in LLM-scale settings. In contrast, adding an exploration bonus (rewarding under-explored responses) is a computationally efficient alternative that implicitly encourages optimistic exploration, i.e., encourages sampling in seldom-visited regions.
>
> **Why is Definition 3.1 an appropriate optimism condition?**: Definition 3.1 defines the condition that the exploratory bonus should satisfy to achieve optimistic exploration. It is directly motivated by the desired behaviour as illustrated in `Figure 1` and `Line 162-167`. The condition in Eq.5 demonstrates that the bonus term should encourage sampling more in seldom sampled (low sampling-probability) regions (small $\pi_s$). The first-order derivative $\frac{\partial\mathcal{L}{\text{bonus}}(r)}{\partial \pi(y|x)}$ represents the gradient that indicates how minimizing the bonus term encourages $\pi(y|x)$. The second-order derivative $\frac{\partial^2 \mathcal{L}{\text{bonus}}(r)}{\partial \pi(y|x) \partial \pi_s(y|x)}$ measures how this encouragement varies with respect to the sampling policy. We require this second-order derivative to be negative to ensure that the bonus strength increases as the sampling probability decreases, i.e., larger encouragement toward less $\pi_s$. This property exactly matches the desired “optimism” behavior illustrated in Figure 1 (III). In contrast, as demonstrated in Theorem 3.3, prior theoretical frameworks have positive second-order derivatives, which means they tend to reinforce already well-explored regions (Fig. 1 (II)). We have supplemented further clarification in `Line 191-200`

---

> ### Author Response · Authors · 2025-11-22
> **Response to Reviewer 3ha8 (2/2)**
>
> > Q2 Most reported improvements (e.g., in Tables 3 and 4) appear to be within 1–2 % over baselines. Could you provide more evidence that these differences are statistically significant and not due to noise or hyperparameter sensitivity?
>
> To demonstrate that our performance gain is significant rather than a luck of hyperparameter or noise, we run three repetitions for DPO and three for each GEB variant, and report their average win rate and standard deviation. The results are provided below and are supplemented in our revised `Appendix F.1`. **The results show that GEB's performance gain is statistically significant**.
>
> Additionally, GEB’s advantages would be most pronounced when optimal behaviors reside in low-$\pi_{\mathrm{ref}}$ regions—i.e., areas underexplored by the reference policy. To demonstrate the problem instances, we include **two additional experiments**.
>
> The first experiment is a toy example of a 1000-arm bandit, where the most preferred action is designed to lie in a region with low reference probability, which means it is unlikely to be sampled during exploration. The results show that the final policy trained with DPO is trapped in local optima, while the policy trained by GEB variants can successfully discover the most preferred action and achieve the global optimum (see `Figure 4`).
>
> The second experiment is on a challenging math reasoning task. We filter out the easy prompts that can be correctly answered by the reference model within two trials. The filter ensures that the correct answers of the remaining prompts lie in a region with less probability, which are more difficult to sample out during exploration. The experiments show that for the Pass@16 performance on three math benchmarks, the model trained by GEB variants can outperform the one trained by DPO with a **substantial performance gap** (**+7%** on Olympiad Bench, **+6%** on AIME 2025).
>
>
> |Method|MATH500 | Olympiad Bench |AIME 2025| Average|
> |---|---|---|---|---|
> |DPO|89.80| 57.78|23.23 |56.94 |
> |GEB-$\pi$| 92.80| 64.59| 28.23 | 61.87  |
> |GEB-$1/\pi$ | 93.00| 65.78|29.48 |  **62.75**  |
> |GEB-$\text{arctanh}\pi$|92.80| 64.59| 29.38 | 62.26 |
>
> We have added these two additional experiments in `Appendix E` and `Line 431-482`.

---

### Official Review · Reviewer_E5mE · 2025-11-07

**Soundness:** 2
**Presentation:** 2
**Contribution:** 3
**Rating:** 6
**Confidence:** 3

**Summary:**

This paper identifies a fundamental theoretical failure in existing exploratory bonus methods for reinforcement learning with human feedback (RLHF) and proposes General Exploratory Bonus (GEB) as a principled solution. The authors prove that current formulations under KL and α-divergence regularization paradoxically bias exploration toward high-probability regions of the reference model rather than uncertain regions, contradicting the "optimism in the face of uncertainty" principle. To address this, they introduce GEB with a reference-dependent reward regulation that provably satisfies optimism, unifies prior heuristic bonuses as special cases, and extends naturally across the α-divergence family. Empirical validation on LLM alignment tasks demonstrates consistent improvements over baselines.

**Strengths:**

1. The failure modes of existing bonuses are rigorously proven, not merely observed. Lemmas 3.1-3.2 and Theorem 3.3 are solid contributions that clarify fundamental issues.

2. The paper provides a general framework (GEB) that unifies prior work and extends beyond KL divergence.

3. Figure 1 provides intuitive understanding, while mathematical sections deliver rigor. Both audiences (practitioners and theorists) can engage.

4. Testing across two LLM backbones, three divergence settings, and both in-domain and out-of-domain tasks demonstrates breadth.

**Weaknesses:**

1. The practical implementation restricts bonus computation to rejected responses for stability, but this isn't theoretically justified. Does this modification preserve the optimism guarantee? If not, the theory's practical applicability is weakened.

2. Win-rate gains are ~1-2% on UltraFeedback and marginal on out-of-domain tasks (Table 4). While consistent, these are not compelling. Concern: Is the gap between theoretical promise and empirical gains expected? Could better tuning of κ yield larger improvements?

3. Missing Ablations: What is the effect of restricting the bonus to rejected responses? Ablating this would isolate theoretical vs. practical design choices.

4. The regret bound in Theorem C.1 has the same dependence on T as prior work. Does GEB offer better constants? This isn't discussed, weakening the theoretical justification for improved sample efficiency.

5. The paper doesn't characterize problem instances where GEB would yield larger improvements. When is exploration in low-pi_{ref}
πref​ regions most critical?

**Questions:**

1. The practical restriction of bonus computation to rejected responses (mentioned in Section 4) lacks theoretical justification. Does this modification affect the optimism guarantee? If so, shouldn't this be analyzed formally?

2. Does GEB improve the constants in the regret bound (Theorem C.1) compared to Cen et al. (2025)? The T-dependence appears identical—is there an improvement in problem-dependent terms?

3. How do the three u designs in Table 2 compare empirically? This would clarify the practical importance of theoretical flexibility.

4. What happens if you remove the restriction to rejected responses and compute the bonus on all responses? This would isolate the theoretical contribution from practical engineering.

5. Are there problem instances or regimes where GEB provides substantially larger improvements than standard methods? Characterizing this would strengthen motivation.

---

> ### Author Response · Authors · 2025-11-22
> **Response to Reviewer E5mE (1/2)**
>
> We thank the reviewer for the thorough comments and suggestions. We deeply appreciate that the reviewer recognized the strengths of our work from various perspectives. Below, we respond to your comments and questions in detail.
>
> > W1/Q1 The practical implementation restricts bonus computation to rejected responses for stability, but this isn't theoretically justified. Does this modification preserve the optimism guarantee?
>
> Thanks for the insightful questions! Restricting the bonus term to rejected responses is a widely-used technique in previous works of exploratory bonus [1][2].
> Meanwhile, the restriction of bonus computation to *rejected responses* is not a theoretical departure. The optimism guarantee in Theorem 4.2 relies on assigning higher probability to low-probability (low ($\pi_{\text{ref}}$)) regions. Since rejected samples precisely correspond to this subset, applying the bonus there **preserves the optimism direction locally**, i.e., gradients still increase ($\pi_\theta(y)$) for underexplored responses.
>
> [1] Shenao Zhang, Donghan Yu, Hiteshi Sharma, Ziyi Yang, Shuohang Wang, Hany Hassan, and Zhaoran Wang. Self-exploring language models: Active preference elicitation for online alignment. CoRR, abs/2405.19332, 2024a.
> [2] Shicong Cen, Jincheng Mei, Katayoon Goshvadi, Hanjun Dai, Tong Yang, Sherry Yang, Dale Schuurmans, Yuejie Chi, and Bo Dai. Value-incentivized preference optimization: A unified approach to online and offline RLHF. 2025.
>
>
> > W3/Q4 What is the effect of restricting the bonus to rejected responses? Ablating this would isolate theoretical vs. practical design choices.
>
> Thanks for the thoughtful suggestion! We compare the win-rate between GEB with rejected responses and with all responses.
>
> |Method|kl w. all|kl w. rejected|hel w. all|hel w. rejected|fkl w. all|fkl w. rejected|
> | --- | --- |---|---| --- |---|---|
> |GEB-$\pi$|79.78 |81.00| 73.40| 75.48|51.32 | 51.68|
> |GEB-$\frac{1}\pi$|79.56 |80.00| 72.25| 73.97|51.61 |52.26|
> |GEB-$\mathrm{arctan}\pi$|79.64 | 79.71|73.11 | 75.69| 51.25| 52.76|
>
> The results show that the widely used technique, i.e., restricting the bonus to rejected responses, is indeed slightly helpful for the performance. The ablation results and discussions are also supplemented in `Appendix F.3`.
>
>
> > W2 Performance gain on UltraFeedback and out-of-domain tasks. Could better tuning of $\kappa$ yield larger improvements?
>
> Thank you for the thoughtful question. We further conduct repetitive runs for the main experiments on UltraFeedback in `Appendix F.1`, which shows that the performance gain of GEB is statistically significant.
>
> As shown in Figure 3, **GEB maintains strong performance across a wide log-scale range of the scaling parameter $\kappa$ (1e-2, 1e-6), indicating low sensitivity and stable gains without extensive tuning.** While more aggressive tuning of $\kappa$ could amplify improvements in specific domains, our results show that GEB already delivers consistent and robust benefits even with minimal hyperparameter search.
>
> In the next response, we clarify and expand the discussion to highlight cases—particularly where GEB’s optimism mechanism can yield more substantial empirical advantages.
>
> > `W5/Q5` The paper doesn't characterize problem instances where GEB would yield larger improvements. When is exploration in low-$\pi_{ref}$ regions most critical?
>
> To demonstrate the problem instances, **we include two additional experiments to show that GEB can yield even stronger performance gains under scenarios where the most preferred/correct action lies in the underexplored areas**.
>
> The first experiment is a toy example of a 1000-arm bandit, where the most preferred action is designed to lie in a region with low reference probability, which means it is unlikely to be sampled during exploration. The results show that the final policy trained with DPO is trapped in local optima, while the policy trained by GEB variants can successfully discover the most preferred action and achieve the global optimum (see `Figure 4`).
>
> The second experiment is on a challenging math reasoning task. We filter out the easy prompts that can be correctly answered by the reference model within two trials. The filter ensures that the correct answers of the remaining prompts lie in a region with less probability, which are more difficult to sample out during exploration. The experiments show that for the Pass@16 performance on three math benchmarks, the model trained by GEB variants can outperform the one trained by DPO with a **substantial performance gap** (**+7%** on Olympiad Bench, **+6%** on AIME 2025).
>
> |Method|MATH500 | Olympiad Bench |AIME 2025| Average|
> |---|---|---|---|---|
> |DPO|89.80| 57.78|23.23 |56.94 |
> |GEB-$\pi$| 92.80| 64.59| 28.23 | 61.87  |
> |GEB-$1/\pi$ | 93.00| 65.78|29.48 |  **62.75**  |
> |GEB-$\text{arctanh}\pi$|92.80| 64.59| 29.38 | 62.26 |
>
> We have added these two additional experiments in `Appendix E` and `Line 431-482`.

---

> ### Author Response · Authors · 2025-11-22
> **Response to Reviewer E5mE (2/2)**
>
> > `W4/Q2` Does GEB offer better regret bound?
>
> GEB only yields an on-par regret bound compared to previous works. However, our work reveals the failure mode of prior theoretical frameworks of optimistic exploration. Moreover, our GEB unifies previous heuristic exploratory bonus and extends the formulation of exploratory bonus to more general $\alpha$-divergence regulated RLHF scenarios.
>
> > `Q3` How do the three $u$ designs in Table 2 compare empirically? This would clarify the practical importance of theoretical flexibility.
>
> It is an insightful question. The three $u$ designs in Table 2 correspond to different instantiations within the GEB framework, each satisfying the optimism condition in Definition 3.1, and show consistent performance gains on the alignment task.
>
> However, the curvature of $u$ will influence the optimization behaviour of the exploratory bonus. For example, when $u=1-\pi+\alpha$, since it is a linear function, the per-trajectory push to shrink $\pi$ is constant. However, if it is a convex function such as $u=1/\pi$, the large $\pi$ will have a smaller derivative. It means the shrinkage would be more conservative compared to the linear case.
>
> **Takeaway**: The curvature of $u$ w.r.t $\pi$ influnces the behavior of exploration bonus. The more convex it is, the more conservative it would be to push the probability mass towards the underexplored region.
>
> We include this discussion in `Appendix F.4`

---

### Official Review · Reviewer_GVVD · 2025-11-07

**Soundness:** 3
**Presentation:** 3
**Contribution:** 2
**Rating:** 6
**Confidence:** 3

**Summary:**

The paper addresses optimistic exploration in reinforcement learning from human feedback (RLHF),  arguing that some existing exploratory bonus methods under KL or $\alpha$-divergence  regularization end up sampling actions with high probability under the reference policy regions  instead of sampling low probability actions. The authors prove theoretically that current  formulations fail to satisfy the optimism principle and introduce the General Exploratory Bonus  (GEB), a framework that introduces reference-dependent reward regulation to counteract  divergence-induced bias. The paper argues that GEB achieves optimism and extends naturally  across the $\alpha$-divergence family. Experiments on large language model alignment tasks  across different divergence settings (KL, Hellinger, forward KL) and model backbones (Llama-3-8B SFT, Mistral-Instruct-v0.3) demonstrate that GEB outperforms baselines.

**Strengths:**

Originality
- The theoretical analysis is insightful, formally proving through Lemmas 3.1-3.2 and Theorem 3.3  that some existing exploratory bonus formulations fail to achieve optimism under KL, $\alpha$- divergence, and general $f$-divergence regularization, with the bonus encouraging sampling from  high probability regions under the reference policy.
- The framework unifies prior heuristic methods (SELM, XPO, VPO) as special cases while extending  beyond the KL-divergence to the $\alpha$-divergence family.

Quality
- The experimental section covers multiple divergence classes, two LLM backbones, both in-domain  and out-of-domain evaluations (AlpacaEval2, MATH-500), with improvements demonstrated  through win-rates, average rewards, and diversity metrics (dist-n scores).

Clarity
- The paper is well-written with clear motivation and effective visualizations (Figure 1).

Significance
- Figure 2 provides evidence that GEB encourages sampling in low-probability reference regions,  validating the theoretical claims.
- GEB is shown to be integrable into iterative online RLHF without additional sampling cost.

**Weaknesses:**

1. Theorem 4.2's optimism condition requires Equation 12 to hold, and $u > \alpha$, but the paper  provides no empirical verification that these conditions hold during training for the proposed $u$ designs.
2. All experiments use only three online iterations following prior work, which may not reveal long term dynamics or potential failure modes.
3. The paper does not report computational overhead relative to baselines - how much wall-clock  time does the additional bonus computation and gradient calculations require?
4. The distinction between the three GEB variants ($u = 1 + \alpha - \pi, 1/\pi, arctanh(1 - \pi) +  \alpha)$ is insufficiently explored - why do they perform differently, and how should practitioners  choose among them?
5. The analysis of $\mathcal{k}$ selection (Figure 3) reveals performance is sensitive to this hyperparameter, with suitable ranges varying across tasks and divergences. Yet, no principled  automatic selection method is provided beyond empirical search.
6. The evaluation focuses exclusively on language model alignment; applicability to other RLHF  domains (robotic control, game playing) remains unexplored.
7. Some of the gains reported in Tables 3 and 4 are marginal. This is possibly a task/training issue. Choosing a domain where the reference model/policy converges on a local suboptimum and GEB performing exploration to converge on the optimal solution might be a better demonstration of GEB empirical success.

**Questions:**

1. How should practitioners choose among the three GEB variants ($u = 1 + \alpha - \pi, 1/\pi, arctanh(1 - \pi) + \alpha$)? Can the authors provide guidance based on task characteristics or on the choice of divergence metric?

2. Have the authors evaluated the performance of GEB beyond language model alignment, such as robotic control or game playing, especially in sparse-reward tasks where optimism-based exploration is generally helpful?

---

> ### Author Response · Authors · 2025-11-22
> **Response to Reviewer GVVD (1/2)**
>
> We thank the reviewer for the positive and constructive comments! We are deeply encouraged that the reviewer recognized the strengths of our work from various perspectives. Below, we respond to your questions in detail.
>
> > W1 Theorem 4.2's optimism condition requires Equation 12 to hold, and $u>\alpha$, but the paper provides no empirical verification that these conditions hold during training for the proposed $u$ designs.
>
> Thank you for the insightful question. We emphasize that the required conditions are **both theoretically satisfied and empirically verified**. As shown in `Lines 355--357`, our proposed designs for $u$ satisfy the condition mentioned in Theorem 4.2 in practice since all designs of $u$ are negatively correlated to $\pi$, and all loss functions in Table 2 empirically maintain the optimism condition $u > \alpha$ throughout training.
>
> To clarify, $\pi(y \mid x)$ (denoted simply as $\pi$ in the discussion) represents the policy’s probability of generating output $y$ given input $x$. Since $0 < \pi(y \mid x) < 1$, and $0 \le \alpha \le 1$, any function $u$ that is negatively correlated with $\pi$ satisfies that condition. The specific forms of $u$ we adopt satisfy these conditions both analytically and empirically:
>
>
> - (1) For $u = 1 + \alpha - \pi$, $u$ is negatively correlated with $\pi$, and we consistently observe $u = 1 + \alpha - \pi > \alpha$ throughout training.
> - (2) For $u = \frac{1}{\pi}$, $u$ is negatively correlated with $\pi$ and empirically satisfies
>     $u = \frac{1}{\pi} > 1 > \alpha$.
> - (3) For $u = \operatorname{arctanh}(1 - \pi) + \alpha$, $u$ is again negatively correlated with $\pi$, and remains above $\alpha$ during optimization.
>
> These observations confirm that our proposed $u$-functions satisfy both the condition set in Theorem 4.2 and the optimism condition $u > \alpha$ not only in theory but also empirically during training.
>
> > W2 Long-term dynamics (beyond three iterations).
>
>
> Thanks for the great suggestion. We adopted the same experimental setting (three iterations) as previous works of exploratory bonus [1][2][3], to ensure fair comparison.
>
> As suggested, we further experimented with 5 iterations of GEB and f-DPO under the KL-divergence, which shows that the performance has reached saturation.
>
> |Method|iteration=3|iteration=4|iteration=5|
> | --- | --- |---|---|
> |f-DPO| 78.42| 78.20 | 78.13|
> |GEB-$\pi$| 81.00| 81.21 |80.78|
> |GEB-$\frac{1}\pi$|80.00|79.71 |79.85|
> |GEB-$\mathrm{arctan}\pi$|79.71| 79.92| 79.56|
>
>
> [1] Shenao Zhang, Donghan Yu, Hiteshi Sharma, Ziyi Yang, Shuohang Wang, Hany Hassan, and Zhaoran Wang. Self-exploring language models: Active preference elicitation for online alignment. CoRR, abs/2405.19332, 2024a.
>
> [2] Tengyang Xie, Dylan J. Foster, Akshay Krishnamurthy, Corby Rosset, Ahmed Awadallah, and Alexander Rakhlin. Exploratory preference optimization: Harnessing implicit q*-approximation for sample-efficient RLHF. ICLR 2025.
>
> [3] Shicong Cen, Jincheng Mei, Katayoon Goshvadi, Hanjun Dai, Tong Yang, Sherry Yang, Dale Schuurmans, Yuejie Chi, and Bo Dai. Value-incentivized preference optimization: A unified approach to online and offline RLHF. 2025.
>
>
> > W3 Computational overhead relative to baselines
>
> The computational complexity of GEB is the same as the baselines. As demonstrated in `Appendix D.1`, the only difference between different baselines and our work is the loss function (Line 5) of Algorithm 1. Thus, our method achieves stronger performance **without incurring additional computational overhead**.
>
>
> >  W4/Q1 Why do three GEB variants perform differently, and how should practitioners choose among them?
>
> It is an insightful question. The three designs in Table 2 correspond to different instantiations within the GEB framework, each satisfying the optimism condition in Definition 3.1, and show consistent performance gains on the alignment task.
>
> However, the curvature of $u$ will influence the optimization behaviour of the exploratory bonus. For example, when $u=1-\pi+\alpha$, since it is a linear function, the per-trajectory push to shrink $\pi$ is constant. However, if it is a convex function such as $u=1/\pi$, the large $\pi$ will have a smaller derivative. It means the shrinkage would be more conservative compared to the linear case.
>
> **Takeaway**: The curvature of $u$ w.r.t $\pi$ influnces the behavior of exploration bonus. The more convex it is, the more conservative it would be to push the probability mass towards the underexplored region.
>
> We include this discussion in `Appendix F.4`

---

> ### Author Response · Authors · 2025-11-22
> **Response to Reviewer GVVD (2/2)**
>
> > W5 Hyperparameter sensitivity.
>
> As shown in Figure 3 and noted in `Lines 484–529`, the performance of GEB remains stable when the selection ratio lies within a broad range (1e-2 to 1e-6). Note that **the $x$-axis is in log scale, so the method is not actually very sensitive to this hyperparameter across this wide interval. Importantly, this wide range (1e-2 to 1e-6) works well across different divergence families**, indicating that GEB is robust to the choice of this hyperparameter in practice.
>
>
>
> > W6/Q2 The evaluation focuses exclusively on language model alignment; applicability to other RLHF domains (robotic control, game playing) remains unexplored.
>
> Thanks for the constructive suggestion. While our scope focuses on language model alignment, the proposed framework is domain-agnostic and can be extended to other RLHF settings, such as robotics and game playing. Our formulation of the bonus term and optimization procedure does not rely on modality-specific assumptions. That said, verifying GEB’s performance in a completely new problem domain—especially in sparse-reward settings—would require an extensive and careful investigation on its own. We view this as an exciting direction for future work and will clarify this generality and potential extension in the revised version.
>
>
> > W7 Choosing a domain where the reference model/policy converges on a local suboptimum and GEB performing exploration to converge on the optimal solution might be a better demonstration of GEB empirical success.
>
> That's a very insightful suggestion. As suggested, we include two additional experiments.
>
> The first experiment is a toy example of a 1000-arm bandit, where the most preferred action is designed to lie in a region of low reference probabilities, which means it is unlikely to be sampled during exploration. The results show that the final policy trained with DPO is trapped in local optima, while the policy trained by GEB variants can successfully discover the most preferred action and achieve the global optimum. See `Figure 4` for details.
>
> The second experiment is on a math reasoning task. We filter out the easy prompts that can be correctly answered by the reference model within two trials. The filter ensures that the correct answers of the remaining prompts lie in a region with less probability, which are more difficult to sample out during exploration. The experiments under the KL divergence show that for the Pass@16 performance on three math benchmarks, the model trained by GEB variants can outperform the one trained by DPO with a consistently noticeable performance gap (**+7%** on Olympiad Bench).
>
> |Method|MATH500 | Olympiad Bench |AIME 2025| Average|
> |---|---|---|---|---|
> |DPO|89.80| 57.78|23.23 |56.94 |
> |GEB-$\pi$| 92.80| 64.59| 28.23 | 61.87  |
> |GEB-$1/\pi$ | 93.00| 65.78|29.48 |  **62.75**  |
> |GEB-$\text{arctanh}\pi$|92.80| 64.59| 29.38 | 62.26 |
>
>
> These two additional experiments (now added in `Appendix E` and `Line 431-482`) show that our GEB variants can make a significant difference under the scenarios where the most preferred / correct action lies in the underexplored areas.

---

### Author Response · Authors · 2025-11-22
**General Response**

## Summary on the recognized strengths of our work
We thank all the reviewers for their time and valuable comments. We are delighted to see that reviewers find our GEB framework **timely**, **relevant** (GVVD, E5mE, 3ha8, e552), **insightful** (GVVD, E5mE, e552), and **comprehensively validated** (GVVD, E5mE). Particularly, several reviewers appreciate that our GEB framework can **unify prior heuristic methods** as special cases while **extending beyond the KL-divergence to the $\alpha$-divergence family** (GVVD, E5mE). Additionally, some reviewers noted that our paper is **clear and well-written** with **rigorous** theoretical proofs (GVVD, E5mE).

As recognized by multiple reviewers, the significance of our work can be summarized as follows:
-  Our work proves that the existing theoretical framework of exploratory bonuses under KL and $\alpha$-divergence regularization fails to achieve optimistic exploration.
- The GEB framework unifies prior heuristic methods (SELM, XPO, VPO) as special cases while extending beyond the KL-divergence to the $\alpha$-divergence family.
-  The experimental section covers multiple divergence classes, two LLM backbones, both in-domain and out-of-domain evaluations (AlpacaEval2, MATH-500), with improvements demonstrated through win-rates, average rewards, and diversity metrics (dist-n scores). Figure 2 provides evidence that GEB encourages sampling in low-probability reference regions, validating the theoretical claims.



## Summary of Manuscript Modification

Based on the reviewers' feedback, we have thoroughly revised our manuscript, with changes highlighted in purple. Below, we summarize the key revisions:
* **Presentation**
  * [3ha8, e552] We enhanced and polished the full text (across the entire paper) and basic notations (`Section 2, line 110-118`) to ensure our revised draft is clear.
  * [3ha8, e552] In revised section 3.1, we provided a rich context on why we need a new condition for optimistic exploration, before introducing Definition 3.1, and further clarified the intuition for the ideal sampling distribution.
  * [e552] We supplement additional background about policy-reparameterized reward and the reasons why we bring up the policy-reparameterized reward.
  * [e552] We revised the statement of Theorem 4.2 to make it grammatically correct.

* **Experiment and Discussion**
  * [e552, 3ha8] We repeated the main experiments in Section 5 multiple times (three runs) for statistical testing and show the improvements are indeed statistically significant in `Appendix F.1`.
  * [E5mE] We added an ablation study on restricting the exploratory bonus to rejected responses in `Appendix F.3` and demonstrated that this restriction, which is widely used in previous work, does help.
  * [E5mE, GVVD] We added additional experiments where the reference model/policy converges on a local suboptimum, and GEB effectively improves exploration and performance in `Appendix E` and `Line 430-482`.
  * [GVVD] We added a discussion between the proposed three GEB variants for practical guidance in `Appendix F.4`, and provided some guidance about how to design $u$.
  * [GVVD] We added discussion on the computational cost of GEB in `Line 1162-1164`, which remains identical to the baselines without overhead.

We believe these revisions have significantly strengthened our manuscript and addressed the key concerns raised.

### Overall Response

In addition, we greatly appreciate the constructive feedback from the reviewers, which further strengthens our work. Altogether, these contributions position GEB as a theoretical framework for advancing optimistic exploration in RLHF, with promising implications for future research.

Below, we address each reviewer’s comments point by point.

---

> ### Comment · Area_Chair_9ZXZ · 2025-11-23
> **Author-Reviewer Discussion**
>
> Dear reviewers,
>
> Please review the authors' response and adjust your rating accordingly. If you have any further questions, please discuss with the authors further.
>
> AC

---

### Meta-Review · Area_Chair_hUWg · 2026-01-06

**Summary:**

The manuscript considers optimistic exploration in RLHF and proposes the general exploratory bonus (GEB) approach to address issues in existing approaches. The approach demonstrates favorable performance in empirical studies.

**Reviewer Concerns:**

The authors address most concerns raised by the reviewers.

**Reviewer Scores:**

I believe the reviewers, especially e552, would increase their score after the full discussion.

---

### Decision · Program_Chairs · 2026-01-26

Accept (Poster)